# CTLA-4 expressing innate lymphoid cells modulate mucosal homeostasis in a microbiota dependent manner

Jonathan W. Lo [1,13], Jan-Hendrik Schroeder [2,13], Luke B. Roberts[2,3,13], Rami Mohamed[2,13], Domenico Cozzetto [1], Gordon Beattie[4,5], Omer S. Omer[2], Ellen M. Ross[6], Frank Heuts[6], Geraldine M. Jowett [7,8,9], Emily Read [7], Matthew Madgwick[1,10,11], Joana F. Neves [7], Tamas Korcsmaros [1,10,11], Richard G. Jenner [12], Lucy S. K. Walker [6], Nick Powell[1] ✉ & Graham M. Lord [2,3] ✉

The maintenance of intestinal homeostasis is a fundamental process critical for organismal integrity. Sitting at the interface of the gut microbiome and mucosal immunity, adaptive and innate lymphoid populations regulate the balance between commensal micro-organisms and pathogens. Checkpoint inhibitors, particularly those targeting the CTLA-4 pathway, disrupt this fine balance and can lead to inflammatory bowel disease and immune checkpoint colitis. Here, we show that CTLA-4 is expressed by innate lymphoid cells and that its expression is regulated by ILC subset-specific cytokine cues in a microbiota-dependent manner. Genetic deletion or antibody blockade of CTLA-4 in multiple in vivo models of colitis demonstrates that this pathway plays a key role in intestinal homeostasis. Lastly, we have found that this observation is conserved in human IBD. We propose that this population of CTLA-4-positive ILC may serve as an important target for the treatment of idiopathic and iatrogenic intestinal inflammation.

Cytotoxic T lymphocyte–associated antigen 4 (CTLA-4 or CD152), is regarded as one of the key inhibitory molecules expressed by conventional CD4[+] and CD8[+] T cells[1–6]. CTLA-4 is homologous to CD28, a costimulatory molecule expressed on the surface of naïve CD4[+] and CD8[+] T cells that promotes T cell activation and proliferation[7,8]. Both CD28 and CTLA-4 bind to the ligands CD80/B7.1 and CD86/B7.2, which are usually found on the surface of antigen presenting cells (APC)[9–11].

CTLA-4 has a 10-100-fold greater affinity for both CD80 and CD86 and outcompetes CD28 binding and can also remove the ligands from cells by a process of transendocytosis[12,13]. Regulatory T cells (T$_{reg}$) suppress aberrant and excessive immune responses via constitutive expression of CTLA-4[14–16]. The importance of this is highlighted in *Ctla4* knockout mice, which die prematurely from multi-organ failure due to excessive immune activation and lymphoproliferation[17,18]. Furthermore, in

[1]Division of Digestive Diseases, Faculty of Medicine, Imperial College London, London, UK. [2]School of Immunology and Microbial Sciences, King's College London, London, UK. [3]Faculty of Biology, Medicine and Health, University of Manchester, Manchester, UK. [4]CRUK City of London Centre Single Cell Genomics Facility, UCL Cancer Institute, University College London, London, UK. [5]Genomics Translational Technology Platform, UCL Cancer Institute, University College London, London, UK. [6]Institute of Immunity & Transplantation, Pears Building, University College London Division of Infection and Immunity, Royal Free Campus, London, UK. [7]Centre for Host-Microbiome Interactions, King's College London, London T, UK. [8]Centre for Craniofacial and Regenerative Biology, King's College London, London, UK. [9]Centre for Stem Cells & Regenerative Medicine, King's College London, London, UK. [10]Earlham Institute, Norwich Research Park, Norwich, UK. [11]Quadram Institute Bioscience, Norwich Research Park, Norwich, UK. [12]UCL Cancer Institute and CRUK City of London Centre, University College London, London, UK. [13]These authors contributed equally: Jonathan W. Lo, Jan-Hendrik Schroeder, Luke B. Roberts, Rami Mohamed. ✉e-mail: nicholas.powell@imperial.ac.uk; graham.lord@manchester.ac.uk

recent years, cancer patients undergoing treatment with anti-CTLA-4 and other immune checkpoint inhibitors (CPI) that block inhibitory molecules and enhance immune activation against tumours, have been found to develop autoimmune and immune-mediated inflammatory diseases[19–27].

The discovery of innate lymphoid cells (ILCs) has been important for the understanding of immune responses, especially at barrier surfaces[28–35]. ILC1 can be activated by stimulation with the cytokines IL-12 and IL-18, express the transcription factor T-bet, and are potent producers of interferon-γ (IFNγ)[28,36]. ILC2 can be activated by alarmin cytokines, including IL-25 and IL-33, highly express the transcription factor GATA3 and are strong producers of interleukin (IL)−5 and IL-13[37–39]. ILC3 are activated by the cytokines IL-23, IL-1β, IL-6, IL-27 and TGF-β and express the transcription factor RORγt but can be further subdivided according to whether they express natural cytotoxicity receptors (NCRs) and the chemokine receptor CCR6. ILC3s are potent producers of IL-22[40,41] and IL-17A[42,43]. Furthermore, CCR6+ embryonic lymphoid tissue inducer (LTi) cells and post-natal LTi-like ILC3s play important roles in lymphoid tissue organogenesis and are potent producers of IL-22[44]. ILCs can also transdifferentiate into other ILC lineages, with parallels to $T_H$ cell plasticity[30,45–52]. CTLA-4 and another inhibitory receptor pair, PD-1 and PD-L1, are expressed by ILC2 during *Nippostrongylus brasiliensis* infection, or stimulation with recombinant IL-25[53]. Transcripts encoding CTLA-4 are increased in intestinal inflammation in both bulk RNA-sequencing and single-cell RNA-sequencing (scRNA-seq) analyses[54–57].

Despite recent discoveries and investigations into the immuno-pathology and biology of ILCs, there are still key limitations in the understanding of these immune cells and how they are regulated. In this study, we aim to further investigate and understand the role of surface checkpoint molecules, in particular CTLA-4, on ILCs and how their expression maintains homeostasis. We show that the checkpoint molecule CTLA-4 is expressed in intestinal lamina propria (LP) ILCs. Furthermore, our data highlight the importance of ILC expression of CTLA-4 for regulating immunological responsiveness in the colon, and identify roles for microbial and inflammatory cues in driving CTLA-4 expression by colonic ILCs.

## Results

### Immune checkpoint molecules are expressed across ILC populations in the colon

We have previously described the importance of ILCs in mediating colitis[30,34,35,51]. Moreover, we have found that ILC1 are a significant driver for the development of DSS-induced colitis and early IFNγ production is likely to be key in the induction of colitis by ILC1[52]. We therefore performed a comprehensive analysis of the expression of well-characterised checkpoint molecules on murine ILCs at single-cell resolution. First, we utilised two publicly available scRNA-seq datasets of FACS-sorted ILCs from the small intestine lamina propria (SI LP) of wild-type (WT) mice[57,58]. The first dataset identified ex-ILC3/ILC1, ILC2, NCR+ ILC3s, LTi-like ILC3 and two undetermined ILC clusters (Fig. 1a). We found that *Ctla4* expression was predominantly detected in ex-ILC3/ILC1, and to a lesser extent in NCR+ ILC3 (Fig. 1b, c). PD-1 (encoded by *Pdcd1*) was not highly expressed in NCR+ ILC3s and ex-ILC3/ILC1s subsets but was detected in ILC2 and LTi-like ILC3 (Fig. 1b, c). PD-L1 and PD-L2 (encoded by *Cd274* and *Pdcd1lg2*, respectively) were detected more widely across all four clusters, but with ILC2 exhibiting the highest expression of PD-L2 (Fig. 1b, c). Interestingly, *Lag3* and VISTA (encoded by *Vsir*) expression was also apparent in LTi-like ILC3, NCR+ ILC3 and ex-ILC3/ILC1 subsets, but less so in ILC2 (Fig. 1b, c). To confirm *Ctla4* expression in NCR+ ILC subsets, we probed a second dataset from Krzywinska et al., which utilised sorted NKp46+ ILC and could, therefore, be used to investigate gene expression by NK cells, as well as ILC1 and NKp46+ ILC3 (Fig. 1d). In agreement with our initial findings, we detected transcripts encoding *Ctla4* in NKp46+ ILC3 and ILC1

populations but not in the NK cells (Fig. 1e, f). *Pdcd1* was not highly expressed in ILC1 and NKp46+ ILC3, but PD-L1 and PD-L2 expression was found across ILC1, NKp46+ ILC3 and NK cells (Fig. 1e, f). The distribution of *Lag3* and *Vsir* expression was consistent with the first dataset, with high expression across NCR+ ILC3 and ILC1 subsets (Fig. 1e, f). Expression of *Havcr2* could only be detected at very low levels across all three ILC subsets (Supplementary Figs. 1a–c).

Blockade of CTLA-4 and PD-1 are linked to checkpoint inhibitor induced colitis in cancer patients[59–61], and genetic variants in *Ctla4* are linked to predisposition to IBD[62,63]. In our recently reported pre-clinical model of immune checkpoint colitis (CPI-C)[64], we identified NCR+ ILC1/3 and ILC2 clusters using scRNA-seq from sorted CD45+ colonic lamina propria cells of healthy wildtype BALB/c mice. Here, we found similar expression of immune checkpoints on ILC subsets, as described above. *Ctla4*, *Lag3*, *Pdcd1lg2* and *Vsir* exhibited higher expression in NCR+ ILC1/3 than ILC2, whereas *Cd274* expression was higher in ILC2 (Supplementary Fig 1d). Thus, results from all these three independent murine scRNA-seq datasets indicate that *Ctla4*, as well as other immune checkpoint molecules, are expressed by certain subsets of ILC.

### CTLA-4 is expressed on certain ILC subsets constitutively and is increased in response to activation and subset-specific cytokine cues

To validate these scRNA-seq data at the protein level, we used flow cytometry to measure CTLA-4 levels in *Rag2−/−* mice (lacking T cells) and, as a negative control, in *Rag2−/− x Ctla4−/−* double knockout mice. Unlike *Ctla4−/−* mice, which develop a spontaneous and fatal phenotype[17,18], *Rag2−/− x Ctla4−/−* mice did not exhibit greater spleen and colon weights compared to *Rag2−/−* mice (Supplementary Fig. 2a and b). CTLA-4 staining was detected in ILC1s, NKp46+ ILC3s and NKp46- ILC3s from *Rag2−/−* mice, while expression of CTLA-4 in ILC2s at steady state was virtually undetectable (Fig. 2a, b).

In PMA and ionomycin stimulated colonic lamina propria (cLP) CD127+ ILCs, from BALB/c mice, there was minimal CTLA-4 expression by ILCs containing Klrg1 (a marker of ILC2) (Fig. 2c). However, in Klrg1- ILCs, we observed CTLA-4 in subsets of both NKp46+ and NKp46- ILCs (Fig. 2c). Furthermore, utilising a C57BL/6 RORγt-eGFP reporter mouse line, we observed co-expression of CTLA-4 in both RORγt-eGFP+ and RORγt-eGFP- ILCs (Fig. 2c), further suggesting that CTLA-4 expression is present among NKp46+ ILC3 and NKp46- CCR6- ILC3. CTLA-4 staining was significantly increased upon stimulation with PMA and ionomycin compared to unstimulated ILC subsets isolated from both BALB/c mice and BALB/c WT animals (Fig. 2d–f). In order to validate the CTLA-4 expression on ILC, we also compared the expression of CTLA-4 on other immune cells that are known to express CTLA-4, like CD4+ T cells, especially $T_{REGS}$, and myeloid cells, such as dendritic cells, using BALB/c wildtype mice (Supplementary Fig. 3a). As expected, we found that the major cell type expressing CTLA-4 was indeed CD4+ T cells, most likely due to including both conventional CD4+ T cells and $T_{REGS}$ within this population. However, our findings of the Ctla-4 expressing ILC1s, NKp46+ ILC3s and NKP46- ILC3s were still valid (Supplementary Figs. 3b and c). These data highlight that CTLA-4 is expressed on ILC subsets and that stimulation of ILCs can increase the expression of CTLA-4 in certain ILC subsets.

To further understand if CTLA-4 could be induced in a subset specific manner, we treated purified cLP ILCs (Supplementary Fig. 4a) with a selection of cytokines and assessed their CTLA-4 levels. Interestingly, we found that a combination of recombinant (r)IL-25 and rIL-33 was a potent inducer of CTLA-4 expression in ILC2 (Fig. 2g). In contrast, CTLA-4 expression by NKp46+ ILC was most strongly induced by rIL-12 + rIL-18 (Fig. 2h). CTLA-4 expression in NKp46+ ILCs and ILC2s was not further enhanced when a cocktail of rIL-1β, rIL-6, rIL-27 and rTGF-β was added to the medium (Fig. 2g, h). These in vitro data indicate that the canonical cytokines used to

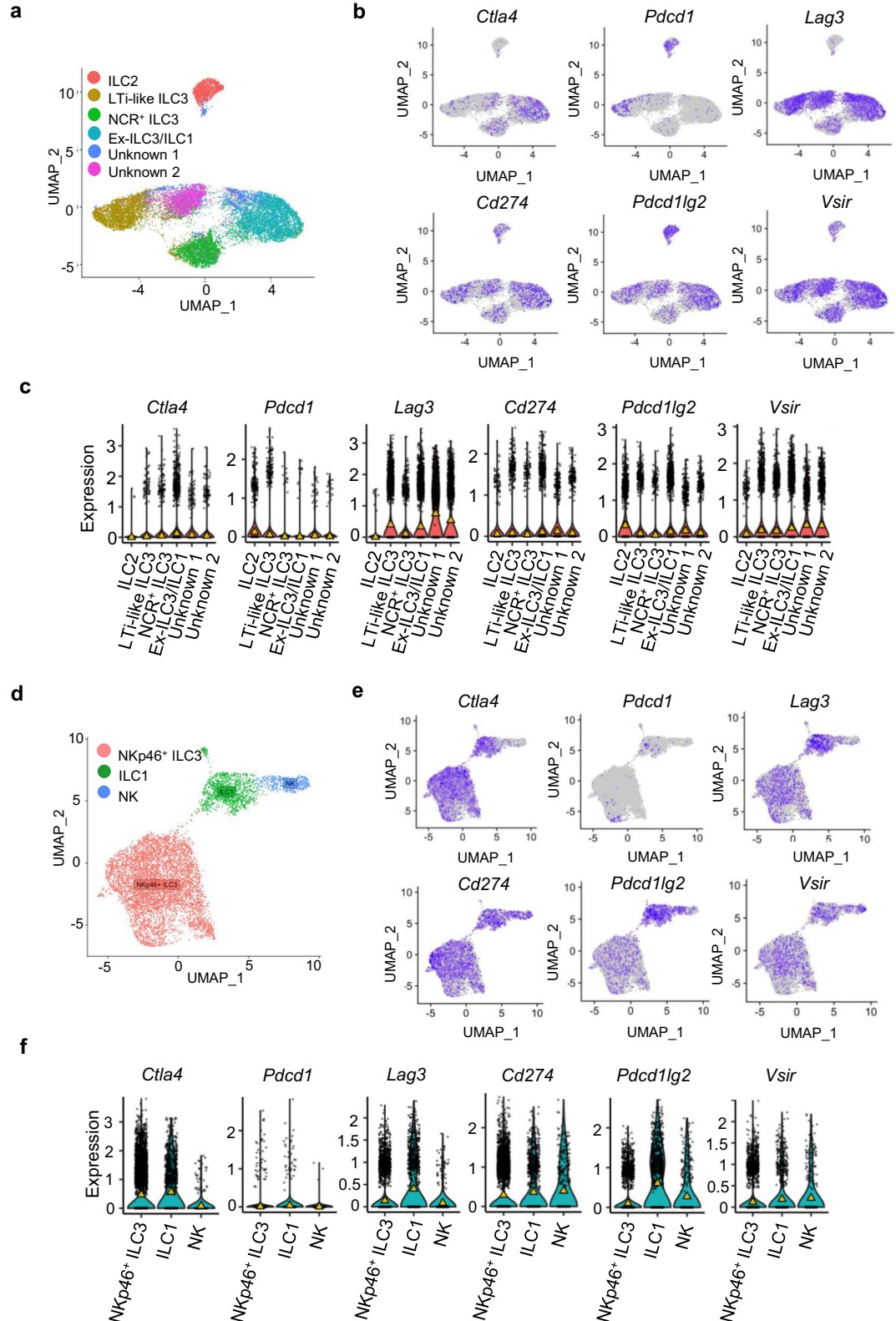

**Fig. 1 | The immune checkpoint transcriptional landscape across intestinal ILC clusters. a** UMAP plot showing the six ILC populations previously identified from CD45+ Lin− CD127+ sorted cells by Fiancette et al. **b** UMAP plots and (**c**) violin plots showing the expression of canonical immune checkpoint inhibitory molecules across the six identified ILC subsets (median shown by the yellow triangle). Cell counts for each clusters are: ILC2 1468, LTi-like ILC3 3736, NCR+ ILC3 2984, Ex-ILC3/ ILC1 5241, Unknown (1) 1115 and Unknown (2) 2412 (**d**) UMAP plot showing the three populations of NKp46+ sorted cells previously identified by Krzywinska et al. **e** UMAP plots and (**f**) violin plots showing the expression of canonical immune checkpoint inhibitory molecules across the three identified ILC populations (median shown by the yellow triangle). Cell counts for the clusters identified here were: NKp46+ ILC3 5739, ILC1 1261 and NK 462.

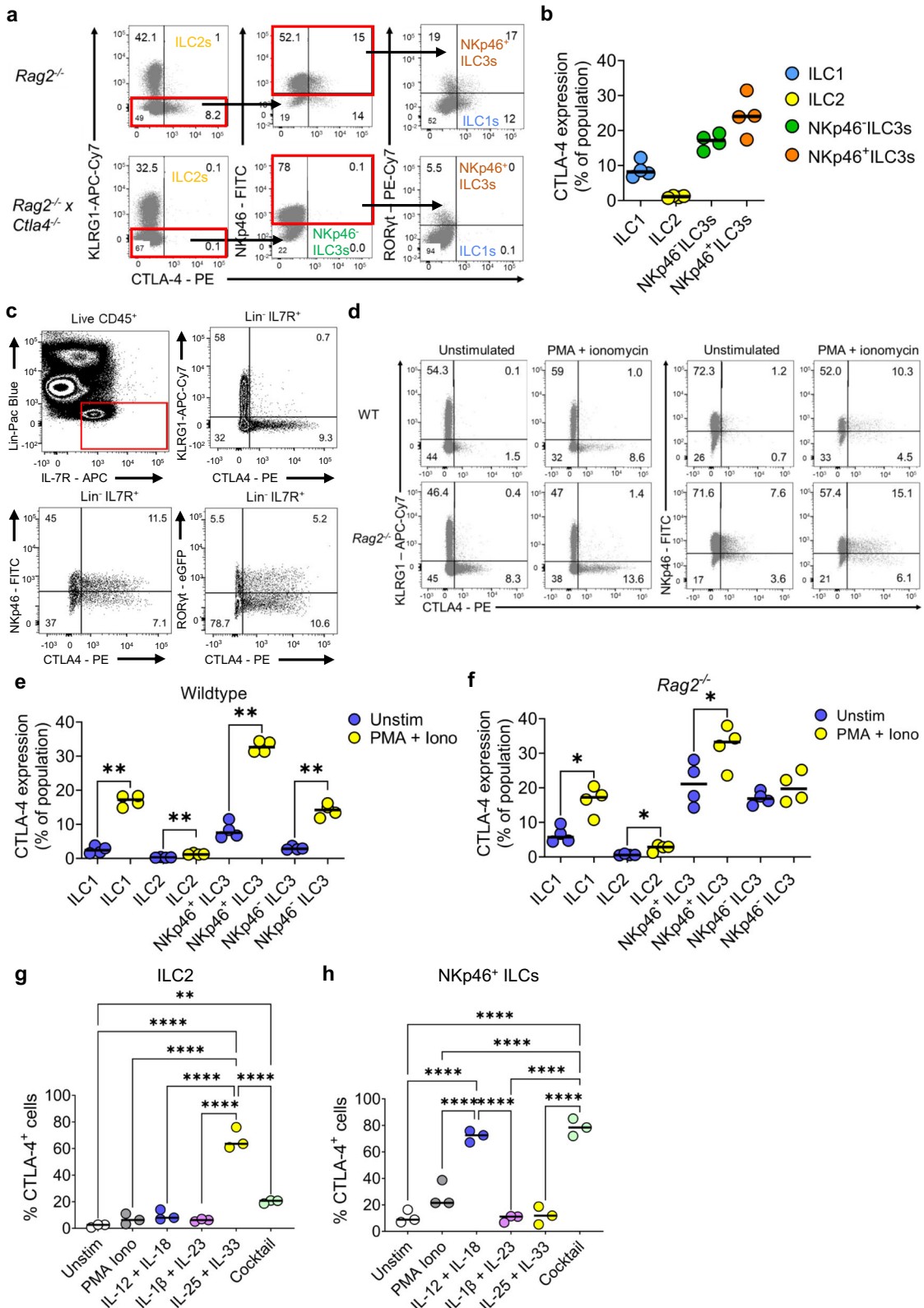

activate the respective ILC2 and NKp46+ ILC subsets can induce the expression of CTLA-4.

An IL-10-producing regulatory ILC subset in the intestine has been previously reported[62], although more recent reports have challenged the validity of this finding[65]. Typically, CTLA-4 expressing Foxp3+ CD4+ T_REG cells produce the anti-inflammatory cytokine IL-10[66]. Therefore, we sought evidence for a population of CD127+ ILC co-expressing IL-10 and CTLA-4 in the cLP and small intestine lamina propria of BALB/c mice. We detected no IL-10-producing ILCs upon stimulation with PMA and ionomycin in the cLP and minimal (4.7%) IL-10-producing ILCs in the small intestine, but none of these co-expressed CTLA-4 (Supplementary Fig. 5a). In contrast, we found a substantial CTLA-4+ ILC population among those stimulated ILCs, but IL-10 and CTLA-4 co-expression was not observed, indicating that CTLA-4 expression is separable from IL-10 production.

**Fig. 2 | CTLA-4 is present on ILC2s and NCR⁺ ILCs. a** Representative flow cytometry plots and (**b**) summary dot plots from BALB/c *Rag2⁻/⁻* (*n* = 4) and BALB/c *Rag2⁻/⁻ x Ctla4⁻/⁻* mice (*n* = 4) showing the proportion of CTLA-4-expressing cLP ILC1, ILC2, NKp46⁺ ILC3s and NKp46⁻ ILC3 cells. **c** Flow cytometry plot showing CTLA-4 in the lamina propria of the colon of BALB/c WT mice and C57BL/6 RORγt-eGFP mice. LPMC were stimulated with PMA, ionomycin and monensin for 4 h. **d** Representative flow cytometry plots and (**e**) summary dot plots from BALB/c WT mice (*n* = 4) and (**f**) BALB/c *Rag2⁻/⁻* mice (*n* = 4) showing the proportion of CTLA-4-positive KLRG1 and NKp46 ILC upon stimulation with PMA and ionomycin. Both e and f used RM one-way ANOVA test with Holm-Sidak's multiple comparison, where

* *P* = 0.0215 and ** *P* = 0.0033 (**g**) KLRG1⁺ (*n* = 3) and (**h**) NKp46⁺ KLRG1⁻ cLP ILC (*n* = 3) were FACS sorted and cultured with OP9-DL1 for 48 hrs in the presence of rIL7, rIL2 and rIL-15 (unstim), with the addition of either PMA and ionomycin, rIL-12 and rIL-18, rIL-1β and rIL-23, rIL-25 and rIL-33, or rIL-12, rIL-18, rIL-1β, rIL-6, rIL-27 and rTGF-β (cocktail). The proportion of CTLA-4-positive KLRG1⁺ CD90.2⁺ and KLRG1⁻ NKp46⁺ CD90.2⁺ cLP ILC was measured upon seeding KLRG1⁺ and KLRG1⁻ cLP ILC, respectively. Both g and h used one-way ANOVA test with Holm-Sidak's multiple comparison * *P* = 0.0111 ** *P* = 0.0015 **** *P* < 0.0001. All experimental n in Fig. 2 are biologically independent mouse samples.

As CD28 and CTLA-4 bind to the same ligands (CD80/B7.1 and CD86/B7.2) and CTLA-4 is known to function by restricting CD28 engagement in T cells, we next investigated whether CD28 was expressed on ILCs at steady state. Analysis of the two publicly available scRNA-seq datasets used in Fig. 1 showed that *Cd28* expression could be detected predominately in the ex-ILC3/ILC1 cluster and a small number of cells in the NCR⁺ ILC3 cluster (Supplementary Fig. 6a, b). In the second dataset by Krzywinska et al., *Cd28* expression could be detected predominately in the ILC1 and NK cluster and a small number of cells in the NKp46⁺ ILC3 cluster (Supplementary Figs. 6c, d). In our own pre-clinical model of immune checkpoint colitis[64], we also found a small number of NCR⁺ ILC1/3 cells which expressed *Cd28* (Supplementary Fig. 6e). Consistent with the scRNA-seq data, we found that, in BALB/c wildtype mice, CD28 was highly expressed on ILC1s and NKp46⁺ ILC3 (Supplementary Fig. 6f). Therefore, ILC-expressed CTLA-4 could potentially alter ILC function by regulating CD28 engagement.

To investigate the role of CD28 on ILCs in mice, we generated a *Rag2⁻/⁻ x Cd28⁻/⁻* mouse line (Supplementary Figs. 7a and b). At steady state, *Rag2⁻/⁻ x Cd28⁻/⁻* mice had a slightly heavier colon, but no other macroscopic differences when compared to *Rag2⁻/⁻Cd28⁺/⁺* littermates (Supplementary Figs. 7c–e). Remarkably, we found that the loss of *Cd28* caused a significant loss of ILC1s, ILC2s and NKp46⁺ ILC3s in the colon (Supplementary Figs. 7f). The phenotype of ILCs was also markedly altered by *Cd28* deficiency, with significantly increased CTLA-4 expression (Supplementary Figs. 8a and b) and IFNγ production (Supplementary Figs. 8c and d) in ILC1s and NKp46⁺ CCR6⁻ ILC3s from the colon. These data indicate that CD28 co-stimulation can modulate ILC number and function, suggesting that CTLA-4 has the potential to alter ILC biology by regulating the CD28 pathway.

### ILCs upregulate CTLA-4 in a microbiota-dependent manner

The composition of the intestinal microbiota has been shown to influence colitis susceptibility in both the general population and cancer patients treated with CPI[67–70]. Therefore, we investigated whether altering the intestinal microbiota would affect CTLA-4 expression in ILCs in the lamina propria of mice. To investigate this, we used C57BL/6 mice maintained under three housing conditions: germ free (GF), specific pathogen-free (SPF), and GF co-housed with SPF mice for 4 weeks prior, in order to recolonise GF mice with SPF microbiota (ex-GF). Altering the microbiota resulted in changes to the numbers of some ILC subsets; ILC1s and ILC2s were increased in GF mice and there was a trend for NKp46⁻ ILC3s to be reduced in ex-GF mice, relative to SPF cell numbers (Supplementary Fig. 9a). Critically, CTLA-4 expression was significantly reduced in ILC1s and both NKp46⁺ and NKp46⁻ ILC3s of GF mice when compared to SPF mice (Fig. 3a, b). Remarkably, the altered CTLA-4 expression in ILC from GF mice was restored in ex-GF mice, indicating that presence of the microbiota actively promotes CTLA-4 expression by colonic ILCs (Fig. 3a, b). Colonic explant biopsies from these mice were cultured for 24 h and analysis of the supernatant by ELISA showed that IL-18 production was significantly reduced in GF mice but restored in ex-GF mice, while IL-12 production was significantly higher in ex-GF mice (Supplementary Fig. 9b). These data highlight the importance of the microbiome in regulating CTLA-4

expression in ILCs and that key drivers of type 1 immunity were also dysregulated by the microbiome.

Next, we tested the impact of the microbiota on the expression of CTLA-4 on ILCs by giving C57BL/6 wild-type mice a cocktail of antibiotics (ampicillin, neomycin, vancomycin and metronidazole) in their drinking water over a period of two weeks. We found that cLP NKp46⁺ IFNγ⁺ ILCs (containing a mixed population of ILC1s and NCR⁺ ILC3s) from the antibiotic-treated mice had a lower per cell expression of CTLA-4 (Supplementary Figs. 9c and d). These data further illustrate that the microbiota play a role in regulating CTLA-4 expression in ILCs.

To further probe the impact of microbial regulation of CTLA-4 expression on ILCs, we transplanted a pro-inflammatory, colitogenic microbiota from TRUC mice[71] to *Rag2⁻/⁻* and *Rag2⁻/⁻ x Ctla4⁻/⁻* by oral gavage. In contrast to *Ctla4* sufficient *Rag2⁻/⁻* mice, where faecal transplantation had no appreciable impact, in *Rag2⁻/⁻ x Ctla4⁻/⁻* double knockout mice transplantation induced features of colitis, including increased colon mass, splenomegaly, increased infiltration of Gr-1^hi neutrophils and expansion of IFNγ producing ILC1s and NKp46⁺ ILC3s (Fig. 3e–g and Supplementary Figs. 10a, 11a–e). There were no major changes in ILC subset numbers post FMT, indicating that it was the enhanced IFNγ production per ILC that was resulting in greater inflammation (Supplementary Figs. 11d and e). We conclude from these data that CTLA-4 is upregulated on ILCs in response to the microbiota and that the composition of the microbiota can induce an inflammatory response when regulation via CTLA-4 on ILCs is perturbed.

### CTLA-4 restricts ILC mediated inflammation in the colon

We next investigated whether inflammation is a driver of immune checkpoint expression on ILCs. We analysed the expression of checkpoint genes using RNA-seq from colonic tissue segments from six different pre-clinical models of colitis, including T cell-mediated models (*Il10⁻/⁻* mice and T cell transfer colitis), chemically induced models (Dinitrobenzene sulphonic acid (DNBS) induced colitis and dextran sulphate sodium (DSS) induced colitis) and two models of ILC-dependent colitis (*Tbx21⁻/⁻ xRag2⁻/⁻* Ulcerative Colitis (TRUC) and anti-CD40 colitis in *Rag⁻/⁻* mice)[35,72]. *Ctla4* was the most highly upregulated checkpoint gene in all the colitis models, with less pronounced changes in the expression of *Pdcd1*, *Lag3*, *Vsir* and *Havcr2* (Fig. 4a). We observed induction of *Ctla4* in the colon in both innate immune mediated models of colitis (Fig. 4a).

To determine whether loss of *Ctla4* resulted in molecular perturbations in the colonic innate immune system, we using RNA-seq to analyse gene expression changes from a colon segment from the distal colon of *Rag2⁻/⁻ Ctla4⁻/⁻* compared to *Rag2⁻/⁻* mice under homeostatic specific pathogen free (SPF) conditions. There were 344 differentially expressed genes (DEGs; FDR < 0.05) in the colon of *Rag2⁻/⁻ x Ctla4⁻/⁻* in comparison with *Rag2⁻/⁻* mice, including 272 upregulated genes and 72 downregulated genes. Notably, many of the most upregulated genes were associated with immune activation (*Ccl7*, *Cxcl10*, *Il18bp*, *Stat2*, *Stat1*) and interferon stimulation (*Gbp3*, *Gbp7*, *Irf7*, *Irf9*, *Ifit1*) (Fig. 4b, c). Pathway analysis demonstrated significant activation (FDR < 0.05) of multiple immune pathways, including interferon signalling (*P* < 4 × 10⁻¹⁰), dendritic cell maturation (*P* < 0.0006) and TREM1 signalling (*P* < 0.007) (Supplementary Fig. 12a). The most strongly

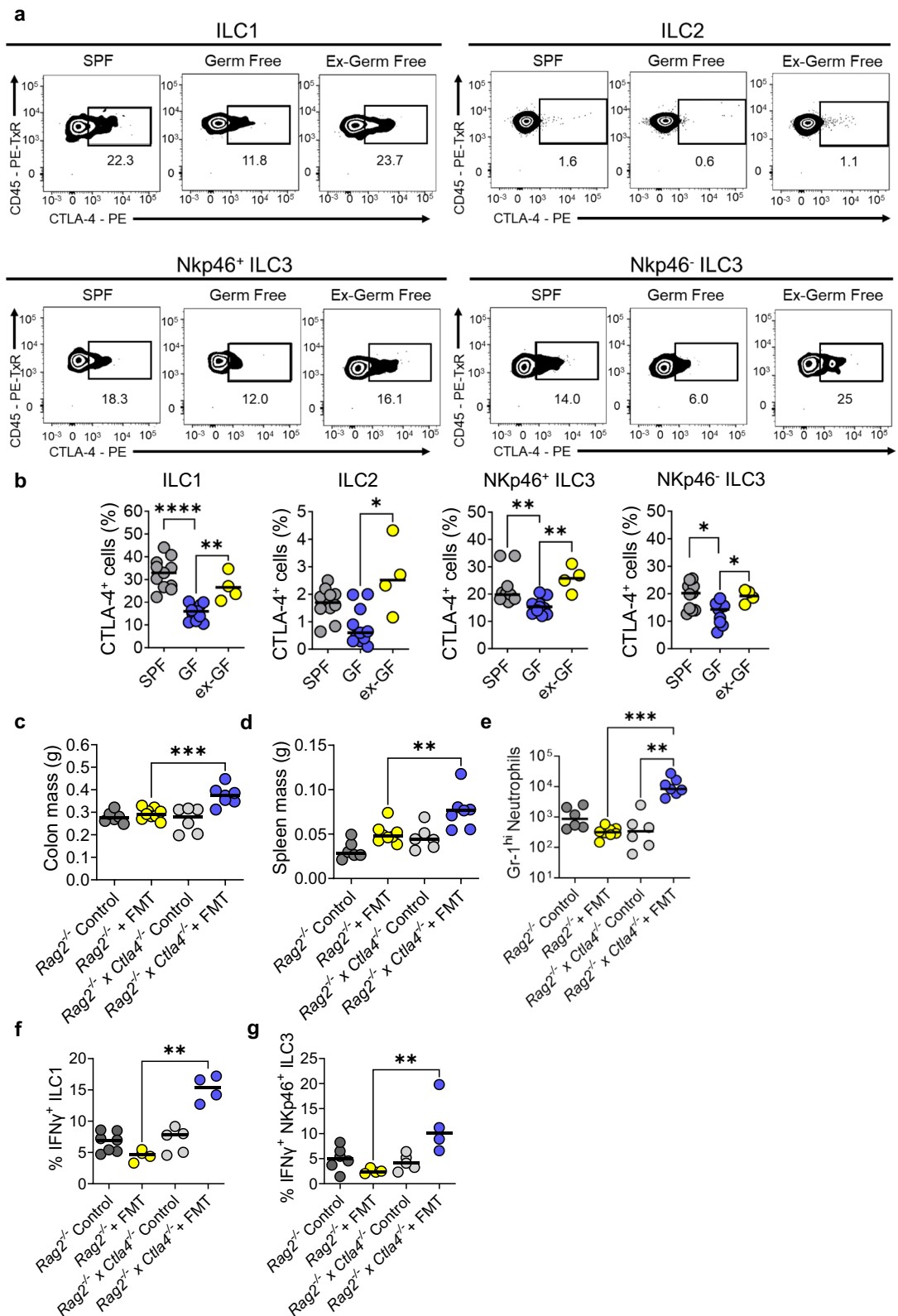

predicted upstream activator of the gene expression changes observed in the colon of *Rag2*[−/−] *Ctla4*[−/−] mice was *Ifng* (*Z*-score= 8.5, $P < 1.1 \times 10^{-50}$) (Supplementary Figs. 12a-b). Other upstream regulators predicted to be activated included *Stat1*, the canonical signalling pathway triggered by *Ifng* (*Z*-score= 7.5, $P < 6.9 \times 10^{-66}$), *Tnf* (*Z*-score=5.5, $P < 2 \times 10^{-22}$) and type 1 interferon pathway components, including *Ifna*, *Ifnar* and *Irf7* (Supplementary Fig. 12a−c). Together,

these data indicate that when *Ctla4* is deficient from innate cells, there is a transcriptional drive towards a more pro-inflammatory IFNγ-mediated transcriptional profile.

We, next, investigated whether this pro-inflammatory gene expression in the innate immune compartment, following genetic deletion or antibody blockade of CTLA-4, had a disease immunological phenotype by using three different colitis models. Previous data

**Fig. 3 | ILC express CTLA-4 in a microbiota-dependent manner. a** Representative flow plots and (**b**) summary statistics showing the percentage of different ILC subsets (ILC1s, ILC2s, NCR⁺ ILC3s and NCR⁻ ILC3s) that are positive for CTLA-4 in specific pathogen free (SPF) (n = 11), germ-free (GF) conditions (n = 11) and GF mice, which have been gavaged with SPF microbiota (ex-GF) (n = 4). One-sided Kruskal-Wallis Test with Dunn's multiple comparison * P = 0.0106 ** P = 0.0048 **** P < 0.0001 (**c**) Colon, (**d**) spleen masses and (**e**) infiltrating Gr-1⁺ neutrophils between control BALB/c *Rag2⁻/⁻* (n = 6), FMT treated BALB/c *Rag2⁻/⁻* (n = 9), control BALB/c *Rag2⁻/⁻ x Ctla4⁻/⁻* (n = 6) or FMT treated BALB/c *Rag2⁻/⁻ x Ctla4⁻/⁻* (n = 7) ** P = 0.0037 *** P = 0.0007 One-sided Kruskal-Wallis Test. **f** IFNγ production from ILC1s (**g**) or NKp46⁺ ILC3s between control BALB/c *Rag2⁻/⁻* (n = 7), FMT treated BALB/c *Rag2⁻/⁻* (n = 4), control BALB/c *Rag2⁻/⁻ x Ctla4⁻/⁻* (n = 5) or FMT treated BALB/c *Rag2⁻/⁻ x Ctla4⁻/⁻* (n = 4) upon restimulation with PMA and ionomycin for 3 h. ** P = 0.0039 for f and P = 0.0052 for g using one-sided Kruskal-Wallis Test with Dunn's multiple comparison. All experimental n in Fig. 3 are biologically independent mouse samples.

showed that the microbiota composition played a role in potentially driving disease, especially in the context of microbiota derived from TRUC mice (Fig. 3c–g). Therefore, we administered a CTLA-4 antagonist to TRUC mice and, when compared to control TRUC mice, we found that they suffered significantly more severe disease outcomes (based on either losing more than 15% weight loss), worse weight loss, significantly worse survival rates, heavier colon and spleen mass, and an increase in infiltrating neutrophils in the colonic lamina propria (Fig. 4d–f and Supplementary Figs. 13a–c). However, despite these clinical differences, there was no increase or changes observed in either IL-17A or IL-13 production from TRUC mice treated with anti-CTLA4 (Supplementary Figs. 13d–e).

*Ctla4* expression was observed in the anti-CD40 model from the bulk RNA-seq data (Fig. 4a). Therefore, we applied the anti-CD40 model to *Rag2⁻/⁻ x Ctla4⁻/⁻* in comparison with *Rag2⁻/⁻* mice to see if there would be a difference in colitis severity. We found that *Rag2⁻/⁻ x Ctla4⁻/⁻* suffered more severe disease, lost more weight, and took longer to recover in this model and had significantly larger spleens, however colon masses and infiltrating neutrophils were the same in comparison to *Rag2⁻/⁻* treated with anti-CD40. (Fig. 4g–j and Supplementary Figs. 14a and b). The only significant difference in cytokine production came from IFNγ⁺ NKp46⁺ ILC3s (Supplementary Fig. 14c). However, *Rag2⁻/⁻* treated with anti-CD40 showed increased expression of CTLA-4 in all subsets of colonic ILCs (Supplementary Fig. 14d).

Lastly, administration of 5% DSS in the drinking water resulted in accelerated weight loss at day 2 and 3 and increased colon mass in *Rag2⁻/⁻ x Ctla4⁻/⁻* mice in comparison with *Rag2⁻/⁻* mice, consistent with more severe colitis in the absence of CTLA-4, but there was no change in spleen mass or colon length (Supplementary Figs. 15a–d). Flow cytometry analysis of cytokines showed a slight increase in IFNγ-producing ILCs in *Rag2⁻/⁻ x Ctla4⁻/⁻* mice in comparison to *Rag2⁻/⁻* mice after treatment with DSS but no change in IL-13/IL-5 producing ILC2 cells (Supplementary Figs. 15e and f).

Taken together, multiple colitis models show that CTLA-4 expressing ILCs play a role in regulating inflammation in the colon.

## CTLA-4⁺ ILC are present in IBD patients

We sought to determine whether our findings in mice translated to patients with IBD. We, therefore, generated a dataset using colon biopsies from patients with active ulcerative colitis (UC) and healthy control subjects without intestinal inflammation to see if *CTLA4* expression was upregulated under inflammatory conditions in the human colon (Supplementary Data 1). RNA-seq data analysis showed that *CTLA4*, and to a lesser extent *PDCD1* and *LAG3*, were all upregulated in the colon of UC patients as compared to healthy control subjects (Fig. 5a). Analysis of a scRNA-seq dataset of human ILCs[73] showed a similar pattern as previous murine scRNA-seq datasets; CTLA-4 and other checkpoint molecule gene expression could be identified on subsets of ILCs, mainly ILC1s and ILC3s, but there was very few cells expressing these in these clusters (Supplementary Figs. 16a-b). To determine whether CTLA-4 was expressed by human colonic ILCs, we extracted lamina propria mononuclear cells of patients undergoing colonoscopy and performed multiparameter flow cytometry. We found that CTLA-4 was present in human ILC1 and NCR⁻ ILC3 cells, when comparing Lineage⁺ IL-7R⁺ shown as a positive control

for CTLA-4 staining on bulk T cells among other immune cells (Fig. 5b and c and Supplementary Fig. 17a). CTLA-4 gMFI levels on ILC1 were also increased in the context of intestinal inflammation in IBD patients compared to healthy controls (Fig. 5d). Interestingly, we were also able to observe CD28 expression on ILC1s and NCR- ILC3s and there was also co-expression with CTLA-4 on these two subsets of ILCs (Supplementary Fig. 17b). These data indicate that CTLA-4 is upregulated on ILCs in IBD.

## ILCs are expanded in, and mediate CPI-induced colitis

In recent years, blockade of immune checkpoints, in particular CTLA-4 and PD-1, has been found to cause a form of IBD, CPI-induced colitis (CPI-C). ILC clusters were identified in our pre-clinical model of CPI-C[64], whereby BALB/c wildtype mice were given the same proinflammatory FMT and anti-CTLA4 and anti-PD1 intra-peritoneal injections to induce CPI-C. However, the role of immune checkpoint blockade in relation to ILCs is not well described in patients with CPI-induced colitis. Since expression of immune checkpoint proteins was observed in ILC populations in wild-type mice (Figs. 1 and 2 and Supplementary Fig. 1), and ILCs are important cells for the pathogenesis of IBD, we hypothesised that there may be a role for ILCs in CPI-induced colitis.

Analysis of the transcriptional changes of sorted live CD45⁺ lymphocytes from the colonic lamina propria (Supplementary Figs. 18a) of wild-type mice treated with combination CPI therapy revealed an increase in the expression of genes associated with interferon signalling, such as *Stat1, Stat4, Gbp5* and *Gbp6*, as well as an increase in the expression of *Ifng* itself, in the cluster identified as NCR⁺ ILC1/3 in mice with CPI-C versus control healthy mice (Fig. 6a, b)[64]. Furthermore, there was an increase in the expression of genes encoding cytotoxic molecules, such as *Gzmb, Gzma* and *Prf1* (Fig. 6a, b). Further analysis at pathway level using the Hallmark database showed increased activity of interferon alpha and interferon gamma pathways in the ILC clusters, indicating an IFNγ-related phenotype (Fig. 6c). Predicted upstream regulators of ILCs in CPI-C included *Tbx21*, encoding the transcription factor T-bet, and genes encoding the cytokines *Il2, Il18* and *Il21*, suggesting that ILC1 may contribute to colitis in this model (Fig. 6d). The retinoic acid receptor (RAR)γ- agonist CD437 and the synthetic retinoid St1926 were also predicted to play a role in the gene expression changes observed in colonic ILC in CPI-induced colitis (Fig. 6d).

To validate this transcriptomic data at the protein level, we measured IFNγ and IL-17A production from IL-7R⁺ CD90⁺ ILCs in BALB/c wildtype mice with CPI-C compared to healthy control mice (for gating see Supplementary Fig. 18b). No differences were seen in cytokine production from ILCs in wild-type mice (Fig. 6e, f). However, in wild-type mice, disease is likely mostly driven by T cells, namely cytotoxic CD8⁺ tissue resident memory T cells[60,61,64]. To scrutinise the role of ILCs in the absence of T cells, we induced CPI-colitis in *Rag2⁻/⁻* mice. Using a similar methodology to our previous wild-type model[64], *Rag2⁻/⁻* mice were treated with the same pro-inflammatory microbiota (FMT, deriving from the TRUC mice) as before and also injected weekly with combination anti-CTLA-4 and anti-PD1 for 3 weeks (Supplementary Fig. 19a). Notably, even in the absence of an adaptive immune system, we still observed induction of disease in *Rag2⁻/⁻* mice, including significantly increased colonic and splenic mass and an increase in the recruitment of neutrophils to the colon (Fig. 6g–i). Overall, the number

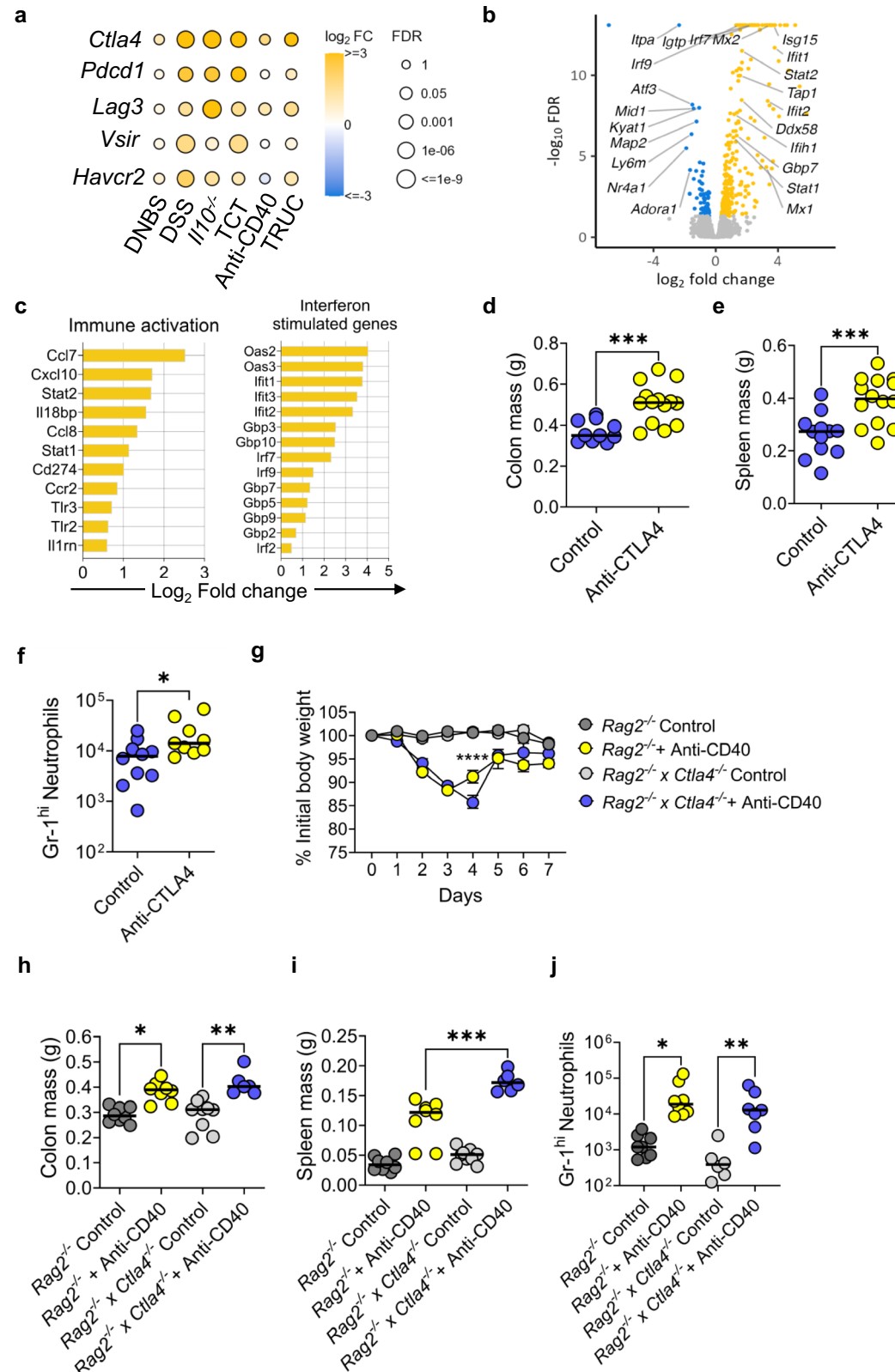

of IFNγ producing ILCs was similar in CPI-C as compared to wildtype mice. However, we did observe a significant increase in polyfunctional IFNγ/IL-17A producing cells in the colon of mice with CPI-C (Fig. 6j, k), consistent with a pathogenic role for these cells[74].

We also interrogated a publicly available scRNA-seq dataset from patients suffering with CPI-C[61]. Initial analysis of the normalized gene expression of *CTLA4* (and other immune checkpoints) did not show

expression of *CTLA4* in the two ILC clusters identified by Luoma et al. (Supplementary Fig. 20a). This was most likely due to the very few cell numbers found in the clusters identified from the bulk CD45+ sorted lymphocytes and also normalizing the gene expression across all clusters, whereby the T cell clusters would have had much higher *CTLA4* gene expression. Therefore, we looked at the actual proportional change in cellular abundance between these two clusters in

**Fig. 4 | CTLA-4 restrains innate immune activation in colitis. a** Heatmap of the changes in immune checkpoint expression in distal colon segments taken from different models of colitis ($Il10^{-/-}$ ($n=3$), DSS ($n=4$), T cell transfer ($n=4$), DNBS ($n=4$), TRUC ($n=4$) and anti-CD40 ($n=4$)) compared to control mice (controls were: WT mice ($n=4$) for $Il10^{-/-}$, DSS and DNBS models; $Rag2^{-/-}$ mice ($n=4$) for T cell transfer, TRUC and anti-CD40 models). **b** Volcano plot showing the gene expression profile of distal colon segments taken from BALB/c $Rag2^{-/-}$ x $Ctla4^{-/-}$ mice ($n=4$) compared to BALB/c $Rag2^{-/-}$ mice ($n=4$). Positive log2 fold-changes indicate upregulation in $Rag2^{-/-}$ x $Ctla4^{-/-}$ mice, while negative log2 fold-changes indicate upregulation in $Rag2^{-/-}$ mice. **c** The most significantly upregulated differentially expressed genes involved in immune activation and interferon-stimulated genes from colon segments taken from $Rag2^{-/-}$ x $Ctla4^{-/-}$ mice ($n=4$) compared to $Rag2^{-/-}$ mice ($n=4$). **d** Colon mass, (**e**) spleen mass and (**f**) infiltrating Gr-1$^+$ neutrophils between TRUC untreated mice (control) ($n=12$) compared to TRUC mice treated with anti-CTLA-4 ($n=14$) * $P=0.035$ *** $P=0.0005$ Two-tailed Mann Whitney U test. **g** weight change (with SEM) $P<0.0001$ 2-way ANOVA Test performed, (**h**) colon mass, (**i**) spleen mass and (**j**) infiltrating Gr-1$^+$ neutrophils between control untreated BALB/c $Rag2^{-/-}$ mice ($n=8$), BALB/c $Rag2^{-/-}$ mice treated with anti-CD40 ($n=8$), untreated BALB/c $Rag2^{-/-}$ x $Ctla4^{-/-}$ mice ($n=9$ for (**h** and **i**), $n=6$ for (**j**)) and BALB/c $Rag2^{-/-}$ x $Ctla4^{-/-}$ mice treated with anti-CD40 ($n=6$ for (**h** and **i**,) $n=7$ for (**j**)) * $P=0.0165$ ** $P=0.0036$ *** $P=0.0005$ One-sided Kruskal-Wallis Test with Dunn's multiple comparison. All experimental $n$ in Fig. 4 are biologically independent mouse samples.

---

healthy control compared to CPI-C patients. We found that the ILC1 population was expanded in patients suffering from CPI-C in comparison to healthy controls and when looking at the transcriptional changes between healthy compared to CPI-C that these ILCs were mainly cytotoxic and had a higher number of genes associated with the interferon signaling pathway (Supplementary Figs. 20b–d). These data suggest that the immunopathology of CPI-C might not solely be driven by adaptive immunity; ILCs may also play a role and could represent a potential therapeutic target in CPI-C too.

## Discussion

This study provides insight into the regulation of CTLA-4 in ILCs, and how blockade or genetic deletion of this crucial regulatory molecule leads to exacerbated disease in mouse models of colitis and in patients with conventional IBD and CPI-C. The barrier surfaces, and the colon in particular, are challenged with maintaining immunological restraint against a multitude of different commensal microbes that vary across individuals and time, whilst remaining poised to repel invading pathogens. In humans and in mice, we identified CTLA-4 expression in colonic ILCs, a population of tissue-resident innate lymphocytes that have been implicated as important early effector cells in host defense against pathogens and restitution of the mucosal barrier, including in conventional inflammatory bowel diseases[75–78]. Our findings suggest that CTLA-4 is an important controller of the host response to the microbiome in the colon. These data are consistent with clinical data from *CTLA4* heterozygote patients who develop life-threatening autoimmunity, including severe intestinal inflammation[79,80].

We show that the ILC subsets predominantly expressing *Ctla4* are ILC1, NCR⁻ ILC3 and NCR⁺ ILC3. When Ctla4 is blocked or deleted, and inflammation is induced, these ILC subsets are responsible for making more proinflammatory cytokine. Notably, *Ctla4* expression in ILCs was increased in $Rag2^{-/-}$ mice lacking adaptive immune cells and was upregulated following cell stimulation. Here, we show that CTLA-4 plays an important role in restraining innate immune activation in the colon in multiple colitis models. Genetic deletion of *Ctla4* in the innate immune compartment triggered a pro-inflammatory transcriptional program in the colon and predisposed to more severe innate-immune mediated colitis, confirming that CTLA-4 curtails ILC-mediated mucosal inflammation.

We also found that *Ctla4* expression was enhanced by ILC activation and regulated by microbial and inflammatory cytokines. Our data support a microbially-driven regulatory circuit of inducible *Ctla4* expression in ILCs, as illustrated by the experimental recolonization of GF mice with SPF microbiota. Previous findings have suggested a reduction in NKp46⁺ ILC3s in GF mice[78,81], however, we found that this was not the case in our study and found an increase in ILC1s and ILC2s in GF mice and no changes in ILC3 subsets. We also showed that the microbiota in mice did impact the expression of CTLA-4 on ILCs as seen when FMT was gavaged in the $Rag2^{-/-}$ mice and $Rag2^{-/-}$ mice lacking the *Ctla4* gene. We propose that the composition of the bacteria within the gut can trigger upregulation of *Ctla4* by ILC, leading in turn to suppression of microbially-driven immune activation.

However, when the CTLA-4-driven regulatory circuit is lost, unopposed mucosal immune activation leads to colitis development. This is consistent with the shifts in the composition of the intestinal microbiota observed in immunotherapy-treated cancer patients who go on to develop CPI-induced colitis[68–70]. Moreover, manipulation of the intestinal microbiota looks to be a promising therapeutic option in the management of CPI-induced colitis[82,83].

We also found that CTLA-4-expressing ILCs were present in patients with IBD. Based on the gene expression changes observed in colonic ILCs in our pre-clinical model of CPI-induced colitis, upstream analysis predicted significant activation of proximal mediators, such as cytokines that induce IFNγ production (*Il18, Il21, Il27* and *Il2*), transcription factors (*Tbx21*), and MAPK kinases (*Map2k1/2*), flagging them as potentially druggable targets. The list of identified drug targets also includes the selective retinoic acid receptor (RAR)γ agonist CD437, and the retinoid ST1926, which were predicted regulators of colonic ILC transcriptomes in CPI-induced colitis. These ILC1 and IFNγ related genes and pathways were also corroborated by a recently published human CPI-induced colitis[61] dataset and could be identified as a potential therapeutically tractable cellular target to this disease.

Lastly, as CTLA-4 and CD28 interact in a mechanistically relevant pathway, we examined CD28 expression on ILCs, revealing that CD28 was predominately expressed on ILC1s. CD28 has previously been noted on human peripheral blood ILCs[84], but has not been well studied in murine ILCs. Mice deficient in both CD80 and CD86 were found to exhibit alterations in ILC subsets in the small intestine[85], consistent with a role for CD28 and/or CTLA-4 on ILCs, however T cell/ILC crosstalk could not be excluded in these lymphoreplete animals. Here, we show in a T cell free setting, loss of CD28 impairs ILC homeostasis and function, potentially contributing to the significantly larger colons found in these $Rag2^{-/-}$ x $Cd28^{-/-}$ mice; most likely due to uncontrolled dysbiosis and inflammation. We further observed expression of CD28 on ILC1s and NCR⁻ ILC3s in patients with IBD, and a proportion of these ILCs co-expressed CTLA-4. Since CD28 is present on ILC subsets, and appears to contribute to their homeostasis and function, it is possible that ILC-expressed CTLA-4 exerts its biological role by regulating CD28 engagement. CD28 signaling in ILC1 could play a significant role to promote the cellularity of ILC1 subsets. This could also explain why CTLA4 appears to restrain ILC1 cellularity in $Rag2^{-/-}$ mice. However, further work would be needed to fully investigate this possibility.

In conclusion, this study demonstrates that the immune checkpoint protein CTLA-4 is expressed by mouse and human colonic ILCs in a manner that is regulated by the microbiota and specific inflammatory cues. Moreover, we show that CTLA-4 plays a key role in restraining innate immune activation during colonic inflammation in a variety of disease settings.

## Methods

### Animal husbandry

C57BL/6 and BALB/c wild-type mice (both Charles River) and BALB/c $Rag2^{-/-}$ mice and BALB/c $Cd28^{-/-}$ mice (Taconic) were sourced

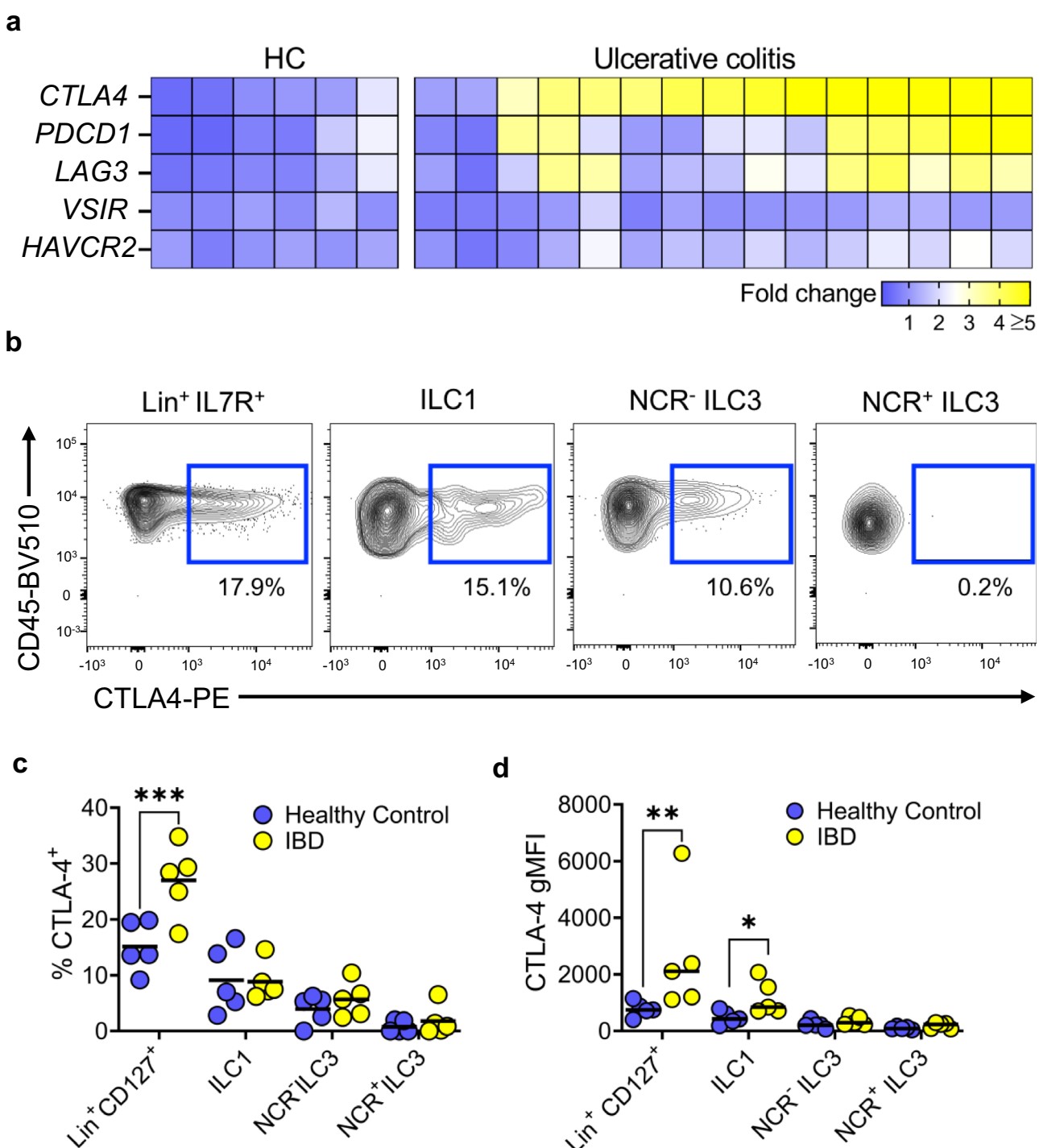

**Fig. 5 | CTLA-4⁺ ILC1 are present in IBD patients. a** Heatmap showing immune checkpoint molecule expression in the sigmoid colon of healthy patients (HC) ($n = 6$) and patients with ulcerative colitis ($n = 15$). **b** Representative flow plots and (**c**) summary statistics showing the percentage of different ILC subsets (ILC1s, NCR⁺ ILC3s and NCR⁻ ILC3s) from patients with IBD ($n = 5$) and healthy controls ($n = 5$) that were positive for CTLA-4 after treatment with PMA, ionomycin and monensin for 3 h. *** $P = 0.0002$ 2-way ANOVA with Šídák's multiple comparisons Test. Lineage⁺ IL-7R⁺ shown as a positive control for CTLA-4 staining on bulk T cells. **d** Dot plot showing the CTLA-4 gMFI in different ILC subsets (ILC1, NCR⁺ ILC3 and NCR⁻ ILC3) in patients with IBD ($n = 5$) compared with healthy controls ($n = 5$) * $P = 0.0423$ ** $P = 0.0036$ 2-way ANOVA with Šídák's multiple comparisons Test. All experimental $n$ in Fig. 5 are biological independent human samples.

commercially. A colony of colitis-free *Rag2⁻/⁻* x *Tbx21⁻/⁻* (TRnUC) mice was generated as described previously. BALB/c *Ctla4⁻/⁻* mice were kindly provided by A. Sharpe (Harvard, Boston) and used to generate *Rag2⁻/⁻* x *Ctla4⁻/⁻* mice. *Rorc*ᴳᶠᴾ mice were a kind gift of Dr Gérard Eberl. BALB/c *Rag2⁻/⁻* mice and BALB/c *Cd28⁻/⁻* mice were crossed to generate the *Rag2⁻/⁻* x *Cd28⁻/⁻* mice. All these mice were housed in specific

pathogen-free facilities at King's College London Biological Services Unit, Imperial College London Central Biomedical Services, University College London Biological Services Unit or Charles River Laboratories. C57BL/6 germ-free mice were housed in germ-free facilities at St. George's University London and also at the University of Manchester. Breeding of *Rag2⁻/⁻* x *Ctla4⁻/⁻* mice, *Rag2⁻/⁻* x *Cd28⁻/⁻* mice and *Rag2⁻/⁻*

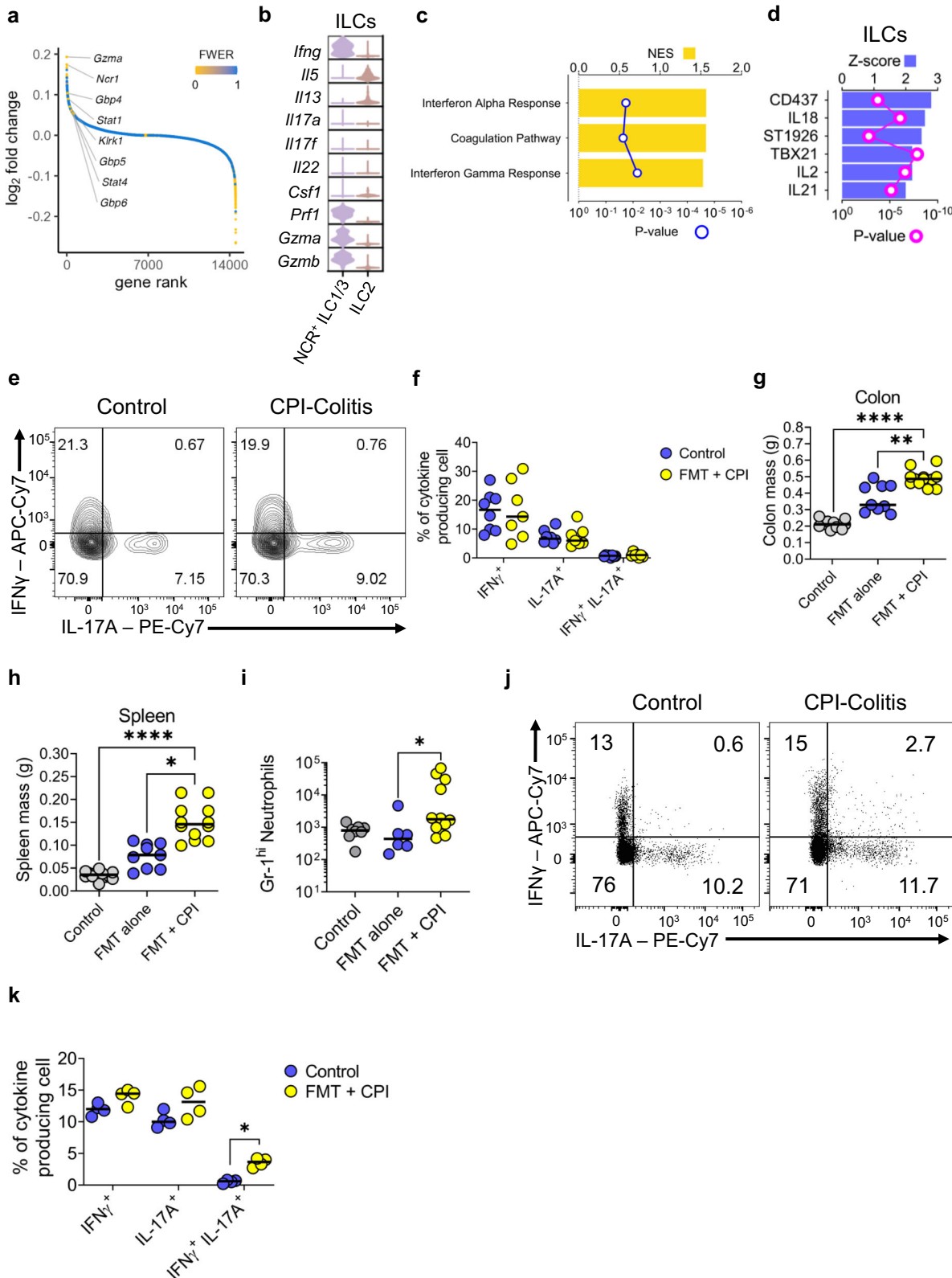

mice was performed at University College London under Home Office Licences PPL PA8A94052 and PP5389651. These mice were housed in individually ventilated cages in a temperature- and humidity-controlled environment with a 14 h light and 10 h dark cycle and ad libitum feeding. Animals were provided with environmental enrichment including cardboard tunnels, paper houses, chewing blocks and aspen wood wool nesting material. All procedures were conducted under licenses (Home Office Licence Numbers PPL: 70/6792, 70/8127, 70/7869, P8999BD42 PA8A94052, PP5389651) from the United Kingdom (UK) Home Office in accordance with The Animals (Scientific Procedures) Act 1986 and licences were approved by each Animal Welfare and Ethical Review Body.

**Fig. 6 | NCR⁺ ILC are expanded in CPI-colitis. a** Distribution of gene expression changes in ILC clusters (ILC2s and NCR⁺ ILC1/3 s) from mice with CPI-induced colitis ($n = 3$) vs those from control wild-type BALB/c mice ($n = 3$). Gene were ranked by decreasing log fold changes and shown as circled coloured according to the associated false discovery rate. **b** Violin plots showing the expression levels of cytokines across the two ILC clusters in wild-type BALB/c control mice ($n = 3$) and mice with CPI-induced colitis ($n = 3$). **c** Pathways, identified by scanning the Hall-mark gene signature dataset using GSEA, significantly upregulated (FDR < 0.05) in the two ILC clusters in mice with CPI-induced colitis ($n = 3$) vs control mice ($n = 3$). **d** Upstream regulators predicted to mediate the gene expression changes in CPI-induced colitis vs control samples from both ILC clusters. **e** Representative flow cytometry plots and (**f**) summary dot plot showing IFNγ and IL-17A cytokine production from Lin⁻ IL-7R⁺ ILCs in wild-type mice treated with CPI-colitis ($n = 7$) compared to untreated mice ($n = 8$). The cells were restimulated with PMA and ionomycin for 3 h prior to analysis. **g** Colon and (**h**) spleen mass in untreated *Rag2⁻/⁻* mice ($n = 12$) and *Rag2⁻/⁻* mice given FMT only ($n = 11$) or FMT + CPI ($n = 12$). * $P = 0.0413$ ** $P = 0.0022$ **** $P < 0.0001$ Kruskal-Wallis Test with Dunn's multiple comparison. **i** Gr-1⁺ neutrophil infiltration between untreated *Rag2⁻/⁻* mice ($n = 8$) and *Rag2⁻/⁻* mice given FMT only ($n = 6$) or FMT + CPI ($n = 12$). * $P = 0.0366$ Kruskal-Wallis Test with Dunn's multiple comparison. **j** Representative flow cytometry plots and (**k**) summary dot plot showing IFNγ and IL-17A cytokine production from Lin⁻ IL-7R⁺ ILCs in *Rag2⁻/⁻* mice treated with CPI-colitis ($n = 4$) compared to untreated control mice ($n = 4$). The cells were restimulated with PMA and ionomycin for 3 h prior to analysis. * $P = 0.0286$ Multiple Mann Whitney U Test. All experimental $n$ in Fig. 6 are biologically independent mouse samples.

## Cell isolation

cLP and Peyer's Patch-free SI LP leucocytes were isolated using a published method[84,85]. Briefly, the epithelium was removed by incubation in HBSS lacking $Mg^{2+}$ or $Ca^{2+}$ (Invitrogen) supplemented with EDTA and HEPES. The tissue was further digested in HBSS supplemented with 2% foetal calf serum (FCS Gold, PAA Laboratories), 0.5 mg/ml collagenase D, 10 μg/ml DNase I and 1.5 mg/ml dispase II (all Roche). The LP lymphocyte-enriched population was harvested from a 40–80% Percoll (GE Healthcare) gradient interface. Leucocytes from human gut were isolated using a published protocol[86].

## Flow cytometry

Flow cytometry was performed as previously described[51,85]. LIVE/DEAD™ stain (ThermoFisher Scientific Inc.) was used to determine cell viability. Lineage cocktails used included antibodies against CD3, CD45R, CD19, CD11b, TER-119, Gr-1, CD5 and FcεRI for murine ILC (ThermoFisher Scientific Inc.) and CD2, CD3, CD14, CD16, CD19, CD235a, and CD56 human ILC (ThermoFisher Scientific Inc.). The FoxP3 staining kit was used for intracellular staining of CTLA-4, transcription factors and cytokines. In case of cytokine analysis, cells were pre-stimulated with 100 ng/ml PMA and 2 μM ionomycin in the presence of 6 μM monensin for either 3 or 4 h prior to FACS analysis as indicated. Samples were acquired using an LSRFortessa™ cell analyzer (Becton Dickinson, USA) and data were analyzed using FlowJo software (Tree Star, USA). Details on the antibodies used in this study are listed in Supplementary Table 1.

## ILC sorting and in vitro culture

Single-cell suspensions from colonic lamina propria were stained with fluorescently labelled antibodies as described and analysed and sorted using a BD FACSAria III cell sorter (BD Biosciences). Antibodies against CD45, lineage markers and IL-7Rα were used to separate CD45⁺ Lin⁻ CD127⁺ cells. ILC were cultured in DMEM supplemented with 10% FCS, 1 x GlutaMax (Gibco), 50 U/ml penicillin, 50 μg/ml streptomycin, 10 mM HEPES, 1x non-essential amino acids (Gibco), 1 mM sodium pyruvate and 50 μM β-mercaptoethanol (Gibco). 20,000 sorted ILCs were plated per well of a flat-bottom 96-well plate pre-coated with 4000 OP9-DL1 following an established method. The medium was further supplemented with rmIL-7 and rhIL-2 (both at 10 ng/ml) and further cultures conditions as indicated. All cytokines were used at a final concentration of 10 ng/ml. Cells were harvested and analysed by flow cytometry after 48 h in culture. In some conditions, the cells were stimulated with 100 ng/ml PMA and 2 μM ionomycin in the presence of 6 μM monensin for 4 h prior to cell harvest.

## Faecal microbiota transplant treatment

Faecal content extracted from the caecum of colitis-confirmed TRUC mice was spun down at 600 g for 10 minutes and reconstituted in sterile PBS with 25% glycerol and then 200 μl was orally gavaged into mice at the beginning of FMT treatment. All mice used for these FMT experiments were 6 weeks of age at the start of experiments. All experimental mice were co-housed in specific pathogen–free facilities at King's College London Biological Services Unit or Imperial Hammersmith CBS. Untreated non-FMT control mice were housed in a separate box to FMT treated mice.

## In vivo murine antibody treatment

Mice used were 6-7 weeks old at the start of experiments and gender matched. Female mice were used in most experiments for this study. Mice treated with immune checkpoint blockade drugs were intraperitoneally (i.p) administered anti-CTLA-4 (9H10, BioXCell), using doses of 200 μg, and anti-PD-1, (RMP1-14, BioXCell) at a dose of 250 μg, once per week[59]. All experimental mice were co-housed in specific pathogen–free facilities at King's College London Biological Services Unit or Imperial Hammersmith CBS. Untreated control mice were housed in a separate box to CPI + FMT treated mice. Mice were injected intraperitoneally with 100 μg anti-CD40, IgG2a monoclonal antibody rat anti-mouse FGK4.5 (BioXCell).

## SPF recolonisation of GF mice and colonic biopsy explant culture

C57BL/6 mice were maintained under three separate housing conditions: germ free (GF), specific pathogen-free (SPF), and GF co-housed with SPF mice for 4 weeks prior, in order to recolonise GF mice with SPF microbiota (ex-GF). Three uniform colon explants, using a 3 mm biopsy punch (Instrapac), from the distal region of each of these mice were taken and cultured for 24 h in complete RPMI in 48well plates @ 37 °C and 5% $CO_2$. After 24 h, supernatants from these wells were collected and spun down to remove any debris and stored at −80°C until required for further analysis.

## ELISA

Cytokine concentrations were measured in culture supernatants by ELISA according to the manufacturers' instructions for IL-12 (EMIL12B) and IL-18 (BMS618-3) (both kits from Thermo Fisher).

## Antibiotic treatment

Cages of mice were treated for 2 weeks with their drinking water supplemented with 2% sucrose, ampicillin (0.5 g/l), vancomycin (0.5 g/l), neomycin (0.5 g/l) and metronidazole (0.5 g/l). Control mice only received 2% sucrose in the drinking water. Mice were sacrificed straight away after this 2-week treatment. Water bottles were monitored to ensure that mice were drinking the water mixture.

## Single-cell RNA-seq

Colonic LPMCs from mice were initially sorted using a FACS Aria machine (BD Biosciences) based on live CD45⁺ gates and taken immediately to the 10X Chromium. Cells were suspended at $1 \times 10^6$/mL in PBS and 10,000 cells were loaded onto the Chromium™ Controller instrument within 15 min after completion of the cell suspension preparation using GemCode Gel Bead and Chip, all from 10x Genomics (Pleasanton, CA), and following the manufacturer's recommendations.

Briefly, cells were partitioned into Gel Beads in Emulsion in the Chromium™ Controller instrument where cell lysis and barcoded reverse transcription of RNA occurred. Libraries were prepared using 10x Genomics Library Kits (3' end V3 kit) and sequenced on an Illumina HiSeq2500 according to the manufacturer's recommendations. Read depth of more than 200 million reads per library, or an approximate average of 20,000 reads per cell was obtained with a recovery of 5000 cells.

The raw 10X Genomics sequencing libraries were processed using the Cell Ranger suite v.3.0.1 to demultiplex base call files, generate single cell feature counts for each library, and finally combine these data into one feature by barcode matrix. Read alignment and gene expression quantification made use of the CellRanger pre-built mouse (mm10 v. 3.0.0) reference data. The individual UMI count matrices were normalised to the same effective sequencing depth before they were aggregated. The merged UMI count matrix was imported in R v.3.6.1 and quality cheques were carried out to mitigate the effects of technical artefacts on downstream analyses. Filtering steps were taken to remove genes detected in less than 3 cells (barcodes), and barcodes with: 1) more than 12,400 UMIs (determined as three median absolute deviations above the median barcode library size); or 2) less than 198 detected genes (determined as three median absolute deviations below the median number of genes for all barcodes after log2 transformation); or 3) expression of the epithelial cell adhesion molecule (*Epcam*) or collagen alpha-1(I) chain (*Col1a1*) that would suggest contamination by epithelial cells or fibroblasts, respectively; or 4) co-expression of genes encoding chains of the CD3 complex (*Cd3d*, *Cd3e* and *Cd3g*) and those of the B cell antigen receptor (*Cd79a*, *Cd79b*) to limit further the impact of multiplets on downstream analyses. The filtered dataset was imported into Seurat v3.1.5[87], and the anchor-based integration workflow was followed to account for biological and technical batch differences between mice and sequencing libraries. For each sample, the UMI counts were normalised using the LogNormalize method and a scale factor equal to 10,000, and the top 10,000 most highly variable genes were identified using the *vst* method. Up to 10,000 integration anchor cells were identified for each pair of count matrices after dimensionality reduction to 20 coordinates via Canonical Component Analysis. These anchor sets guided the iterative process integrating the individual sample data into one shared space with all genes passing initial quality cheques. The expression data were scaled after regressing out the following sources of biological and technical variation: mouse id, sequencing library preparation and sequencing batches, number of UMIs detected for each cell and percent of UMIs for mitochondrial genes in each cell. After running Principal Component Analysis of the integrated expression data and inspecting the scree plot, 30 coordinates were used to represent the whole dataset into two dimensions using tSNE.

To cluster the integrated dataset, the k = 20 nearest neighbours of each cell were determined based the Euclidean distance of their expression profiles projected onto the first 30 principal components previously identified. A shared nearest neighbour graph was then built from these data to represent the neighbourhood overlap between pairs of cells using the Jaccard similarity index. From this graph, 24 cell clusters (identified with integers from 0 to 23) were computed by the Louvain algorithm for modularity optimisation with a resolution parameter equal to 0.8. Markers for each cluster were identified by differential expression analysis using MAST[88]. Only genes expressed in at least 5% of the cells in the cluster under consideration or the rest of the cell population were tested. Genes were considered differentially expressed if the absolute value of the natural log fold change was greater than 0.25 and the Bonferroni-adjusted *P value* was less than 0.001.

Clusters were annotated to broad cell types using the SingleR (v.1.0.6) package[89] and the Immunologic Genome Project (ImmGen) transcriptomic datasets for sorted populations of mouse immune cells[90]. To this end, SingleR was used to calculate cluster-level gene expression profiles from the individual cell's data, and then to classify them using its correlation-based iterative algorithm with default parameter settings. The classification process yielded confident assignments based on the pruned scores for all but two sub-populations.

## Analysis of publicly available datasets

Single cell count matrices were obtained from ArrayExpress accessions E-MTAB-9795[57], E-MTAB-11238[58] and GEO accession GSE150050[73]. Count matrices were normalised using Seurat's NormalizeData function before visualisation. UMAP co-ordinates and clustering metadata were as originally published. Therefore, downstream processing steps can be considered identical to those carried out as previously published. Each matrix was then normalised using SCTransform, followed by RunPCA (PCs=30) and RunUMAP (dims=30). Shared nearest neighbour and clustering were carried out using FindNeighbours (dims=30) and FindClusters respectively. NormalizeData was then ran, and this assay was used for downstream visualisation and differential expression analysis using the MAST algorithm[88].

Processed data from droplet-based scRNA-seq profiles of CD45+ immune cells from healthy donors and patients with CPI colitis were downloaded from NCBI GEO (Series GSE144469)[61]. The UMI count matrices were filtered and normalised as described in the original publication using Seurat. The resulting dataset was integrated following the reciprocal PCA workflow and expression data were finally scaled by regressing out the percentage of mitochondrial transcripts and the number of UMIs detected in each cell. Cell-type assignments and UMAP coordinates were kindly provided by the authors. Average normalised gene expression for each population in CPI colitis was calculated and compared to equivalent data from the mouse data presented here by means of Spearman correlation. These expression profiles were also scanned for enrichment of hallmark signatures using single-sample GSEA. Spearman correlation between the output enrichment scores from the mouse and human data was also estimated.

## Bulk RNA-seq analysis

For bulk RNAseq analysis, RNA was extracted from distal colon segments and purified cells with Qiazol reagent (Invitrogen). In this method, distilled solutions of RNA were purified using chloroform and isopropanol and eluted with ethanol before being resuspended in RNAse-free water. RNA was cleaned up using the Qiagen RNeasy Microkit and performing the DNA clean-up protocol as listed in the manufacturer's guidelines. RNA samples were then checked for quality, contamination and concentration using a NanoDrop, Qubit spectrophotometer and Bioanalyzer. RNA with only a RIN score of above 7 were used for RNA sequencing. RNA was then stored at -80 °C prior to further analysis.

The quality of the raw library files was inspected with fastQC. Raw reads were trimmed and filtered to remove adaptor contamination and poor-quality bases using trimmomatic. Depending on the source organism, the resulting read files were mapped to the GRCm38 (mouse) genome assembly using Hisat2 with default parameters. The number of reads mapping to the genomic features annotated in Ensembl with a MAPQ score higher than or equal to 10 was calculated for all samples using htseq-count with default parameters.

Differential expression analysis between sample groups was performed in R using the Wald test as implemented in the DESeq2 package. P-values were adjusted for multiple testing according to the Benjamini and Hochberg procedure. The P- and corresponding FDR values were re-estimated empirically with fdrtool, when the histograms of the initial P-value distributions showed that the assumptions of the test were not met.

## Pre-clinical colitis models

Dextran sulphate sodium (DSS) (36-50 Kda, MP Biomedicals, Ontario, USA) colitis was induced in mice by adding 3% DSS for wild-type model and 5% for *Rag2*[-/-] mice to the drinking water for a period of 5 days, after which DSS was removed. Mice were culled after 5 days of being on normal water. For the *Rag2*[-/-] x *Ctla4*[-/-] mouse related experiments, the relevant mice were sacrificed 2 days after the beginning of the treatment with 5% DSS in order to isolate cLP leucocytes. Anti-CD40 agonistic antibody induced colitis in *Rag2*[-/-] mice following i.p. injection of 100 μg of rat anti-mouse (clone: FGK4.5) (BioXCell, West Lebanon, USA). T cell transfer colitis was induced by injecting (i.p.) FACs sorted 0.5×10[6] live naïve CD4[+]CD25[-]CD62L[+]CD44[-] splenic T cells into *Rag2*[-/-] mice[85]. DNBS colitis was induced in C57BL/6 wild-type mice using DNBS (Sigma-Aldrich, France) resuspended in 50 μl of 30% ethanol (EtOH) in PBS. The DNBS solution was administered on day 1 by injection with a tuberculin syringe (Terumo, France) and a flexible plastic probe inserted 3.5 cm into the colon. TRUC colitis has been described elsewhere[35,71]. Immune checkpoint colitis was induced using our previously published method[64], in brief, mice were given the FMT and combination anti-CTLA-4 and anti-PD-1, as described above, for 3 weeks.

## Human samples

Studies in human tissues received ethical approval from Guy's and St Thomas's Trust (REC number: 15/LO/1998). Typically, 12-16 biopsies were taken from each patient. 1 of these biopsies would be taken in RNAlater for RNA extraction and bulk RNA-seq using the same above-described method. Colonic lamina propria mononuclear cells (cLPMCs) were isolated using either grids or by the digestion protocol described above from endoscopically acquired biopsy specimens. Isolated cLPMCs were stimulated with 100 ng/ml PMA and 2 μM ionomycin in the presence of 6 μM monensin for 3 h and then stained for flow cytometry as described above. Meta-data for patients used in this study are attached to Supplementary Data 1.

## Statistics

Results are expressed as median ± IQR or mean ± SEM. Data were analysed using Two-way Student's t-test, Two-way Mann-Whitney U test, Two-Way ANOVA Kruskal-Wallis test, as appropriate, using GraphPad Prism 10.0 (GraphPad Inc., USA). No data were excluded from the analyses. * $P < 0.05$; ** $P < 0.01$; *** $P < 0.001$; **** $P < 0.0001$.

## Reporting summary

Further information on research design is available in the Nature Portfolio Reporting Summary linked to this article.

## Data availability

All raw data in this study are available in the Source Data file, which has been uploaded with this manuscript. NGS data presented in the manuscript have been made publicly available through the Gene Expression Omnibus (GEO) database found at https://www.ncbi.nlm.nih.gov/geo/. GSE208395 "Transcriptome profiling of colon biopsies from pre-clinical models of colitis" for the bulk RNA-seq comparing different pre-clinical mouse models used for Fig. 4a (https://www.ncbi.nlm.nih.gov/geo/query/acc.cgi?acc=GSE208395). GSE254247 for Fig. 4b-c comparing colons of *Rag2*[-/-] against *Rag2*[-/-] x *Ctla4*[-/-] titled "Transcriptomic profiling of the role of Ctla4 in the innate immune system" (https://www.ncbi.nlm.nih.gov/geo/query/acc.cgi?acc=GSE254247). GSE224758 for Fig. 5a comparing healthy controls with patients with ulcerative colitis titled "Gene expression profiling of colon biopsies from ulcerative colitis patients and healthy volunteers" (https://www.ncbi.nlm.nih.gov/geo/query/acc.cgi?acc=GSE224758). GSE222959 titled "Single cell transcriptomics reveals colonic lymphocyte remodelling and emergence of polyfunctional, cytolytic lymphocyte responses in CPI-induced colitis" for the single cell RNA-seq

data comparing the wildtype CPI + FMT treated BALB/c CPI-C mice to healthy untreated wildtype BALB/c mice used in Fig. 6 (https://www.ncbi.nlm.nih.gov/geo/query/acc.cgi?acc=GSE222959) and has previously been published[64]. Source data are provided with this paper.

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

## Acknowledgements

We would like to acknowledge the BRC FlowCore at King's College London and the MRC LMS FlowCore facility at Imperial College London. The views expressed are those of the author(s) and not necessarily those of the NHS, the NIHR, Imperial College London, King's College London, or the Department of Health and Social Care. We acknowledge Novogene for the RNA sequencing and the BRC Genomics Facility at King's College London for the 10X Genomics single-cell RNA sequencing. We would like to thank Dr Gérard Eberl (Institut Pasteur, Paris) for providing *Rorc*^GFP mice. We also would like to thank Professor Mike Curtis (King's College London), Rene Ocho, Emma Mustafa and Robert Bond (all St. George's University London) and Matthew Hepworth (University of Manchester) for generating and providing germ-free mice. We also thank Professor Zúñiga-Pflücker (Sunnybrook Research Institute, University of Toronto) for contributing OP9-DL1 cells. This study was supported by grants awarded by the Wellcome Trust (NP, WT101159 and WT225875, LSKW, 220772/Z/20/Z) and the UKRI Medical Research Council (GL and RGJ, MR/M003493/1 and MR/R001413/1; GL, MR/K002996/1; LSKW, MR/S0091401/1). NP was also funded by and the Imperial National Institute for Health Research (NIHR) Biomedical Research Centre (BRC). JFN acknowledges an RCUK/UKRI Rutherford Fund fellowship (MR/R024812/1), and GMJ (203757/Z/16/A) and ER (215027/Z/18/Z) were funded by a PhD fellowship from the Wellcome Trust. The research was also supported by the National Institute for Health Research (NIHR) Biomedical Research Centre at Guy's and St Thomas and King's College London (GL). The work of TK was supported by the UKRI BBSRC Gut Microbes and Health Institute Strategic Programme (BB/R012490/1 and its constituent projects BBS/E/F/000PR10353 and BBS/E/F/000PR10355) as well as a BBSRC Core Strategic Programme Grant for Genomes to Food Security (BB/CSP1720/1 and its constituent work packages, BBS/E/T/000PR9819 and BBS/E/T/000PR9817). TK was also supported by the UKRI BBSRC Institute Strategic Programme on Food Microbiome and Health BB/X011054/1 and its constituent project BBS/E/F/000PR13631. The views expressed are those of the author(s) and not necessarily those of the NHS, the NIHR, or the Department of Health. Work at the CRUK City of London Centre Single Cell Genomics Facility and Cancer Institute Genomics Translational Technology Platform was supported by the CRUK City of London Centre Award [C7893/A26233].

## Author contributions

Study concept and design: G.M.L., N.P., and L.S.K.W. Acquisition of data: J.W.L., J.H.S., L.B.R., R.M., D.C., G.B., O.S.O., and M.M. Data analysis and interpretation: J.W.L., J.H.S., L.B.R., R.M., D.C., G.B., O.S.O., and M.M. Technical support: E.M.R., F.H., G.M.J. and E.R. Patient recruitment and consenting: O.S.O. and N.P. Obtained funding: G.M.L., L.S.K.W., N.P., J.F.N., R.G.J., T.K and R.M. Drafting of manuscript: J.W.L., J.H.S and R.M. Edits to manuscript: L.B.R., L.S.K.W., N.P., and G.M.L. Study supervision: G.M.L., and N.P.

## Competing interests

The authors declare no competing interests.
