## [Peer Review File · Nature Communications]

CTLA-4 expressing innate lymphoid cells modulate mucosal homeostasis in a microbiota dependent mannerReviewers' Comments:

Reviewer #1:

Remarks to the Author:

Lo and colleagues report here that innate lymphoid cells (ILC) express the cytotoxic T lymphocyte-associated antigen 4 (CTLA-4) checkpoint with predominant expression in ILC1 and ILC3 regulated by cytokine cues and the microbiota. The genetic deletion of CTLA-4 in mice or blocking CTLA-4 with antibodies indicated the relevance of CTLA-4 expressed by ILCs for maintaining the integrity of intestinal homeostasis in chemical and immune checkpoint inhibitor (CPI) colitis models. These results are interesting considering these results and the clinical relevance of CPI colitis in tumor patients treated with checkpoint inhibitors.

Previous reports (PMID: 9430233) showed that the lymphoproliferative disorder reported in *Ctla4*^{-/-} mice is mediated by CD4 T cells as indicated in experiments with depletion of CD4 T cells in *Ctla4*^{-/-} animals. The reported difference in the percent of initial body weight loss between *Rag2*^{-/-} and *Rag2*^{-/-} x *Ctla4*^{-/-} challenged with Dextran Sodium Sulfate (DSS) is minor and not significant except on day 2. These findings question the relevance of ILC2 expressing CTLA-4 for developing the lymphoproliferative disorders observed in *CTLA-4*^{-/-} mice and potentially their relevance to CPI colitis.

The authors report that the intestinal microbiota or the stimulation with a cytokine cocktail (IL-12 + IL-18 for Nkp46⁺ ILCs) induces *Ctla4* expression by ILCs as indicated by experiments with antibiotics-treated mice, germ-free animals, or in vitro stimulation. Information on how these different findings are linked to driving *Ctla4* expression by ILCs must be provided.

The authors need to suggest mechanisms on how CTLA-4 restricts the suggested immunophenotype mediated by ILCs in DSS and CPI colitis. The manuscript focuses on exacerbated IFN γ production, but the manuscript does not touch other potential intrinsic and extrinsic pathways, such as ligand competition, inhibitory signaling, or modulating the function of antigen-presenting cells by reverse signaling.

Moreover, the methods could be more precisely described. Mainly the CPI colitis model needs to be explained. The authors have submitted a second manuscript describing the CPI colitis model. Still, I need access to this manuscript to explain the CPI model in detail to review its significance of the CPI model for this manuscript.

Major points

1. Page 6, lines 126 to 129, describing figure 1, panel D. It remains to be seen in the text that these results have been derived from the Krzywinska data set. Moreover, figure 1 focuses on small intestinal *Ctla4* expression by ILCs, whereas most of the manuscript focuses on the large intestine. Authors could, for example, also analyze the GSE150050 datasets to show CTLA-4 expression by human colonic ILCs.

2. Presenting the representative flow cytometry plots for all analyzed ILC subtypes is more interesting. Consider moving the dot plots of the supplementary figure 2, panel A, to the central figure. Figure 2, panels A and B, the *Rag2*^{-/-} x *Ctla4*^{-/-} animals serve as a control. I do not think that in all summary plots, the negative control has to be presented. The corresponding figure legends do not describe what population was gated for the representative dot plots of panel A.

3. In the figure legend of Figure 2, panel C, please state that the WT animals are BALB/c mice. On page 8, line 157, please explain why BALB/c mice have been used as WT animals. Add the C57BL/6 WT controls to the panels with the Rag2^{-/-} and RORγt-eGFP mice.

4. Figure 2, panels C to H, there is also the possibility that the Ctla-4 homolog CD28 is upregulated by PMA / Iono or cytokine cocktail stimulation. It might also mean that activated ILCs and not resting ILCs express Ctla-4, which could be further emphasized by detecting other activation markers.

5. Please indicate on page 9, line 193, that WT animals are C57BL/6 mice.

6. The presented results of Figure 3 show that ILCs of antibiotic-treated and germ-free animals have reduced Ctla-4 expression. Since the literature is controversial if the number of ILCs is altered in germ-free and antibiotic-treated animals, indicate whether in these animals the ILC numbers are reduced or not changed. In addition, colonize germ-free animals with a microbiota to show that the microbiota upregulates Ctla-4, as stated on page 10, line 208. In the same experiment, please measure IL-12 and IL-18 to learn whether IL-12 and IL-18 mediate the effects of the microbiota on Ctla-4 expression by ILCs.

7. Since the reported differences in the percent of initial body weight loss between Rag2^{-/-} and Rag2^{-/-} x Ctla4^{-/-} exposed to Dextran Sodium Sulfate (DSS) are small, additional read-out parameters, such as histology, disease activity scores, and colon length, are warranted.

8. Figure 5, the presented heat map of an RNA-seq experiment of panel A, remains somewhat unclear if the submitted data are obtained from the re-analysis of a previous data set or whether this data stems from a newly generated data set. The IBD characteristics, including disease activity and course and immunosuppressive therapies, from which the material for the presented RNA-seq and flow cytometry data stems, need to be given. Also, the authors may consider offering Ctla4 expression from the non-inflamed regions of the same patients to exclude the possibility that individual differences and batch effects may have contributed to the reported results.

9. In Figure 6, panels F to J, the 'FMT' is not explained in the text describing the figure. The authors may also consider re-analyzing the GSE144469 data set for CTLA-4 expression by colonic ILCs in addition to the presented flow cytometry data from the peripheral blood.

10. Methods

- Animal husbandry: The animal protocol number of the local animal welfare committee missing
- Colitis models: Have littermates been used?
- Flow cytometry: clone numbers of the used antibodies missing
- Antibiotic treatment: Since the antibiotics have been given in the drinking water, each animal's dose is difficult to judge.
- Immune checkpoint colitis not described.

Minor

1. The clusters are not annotated in the UMAP graph, panel A, supplementary figure 1.
2. Flow cytometry dot blots: The manuscript does not give x- and y-axis scales.

3. On page 8, line 165, correct the spelling mistake “non-soecific”.

Reviewer #2:

Remarks to the Author:

In this manuscript, Lo and colleagues describe the induction of CTLA4 on ILC1/3 driven by gut microbial and inflammatory cues and its involvement in the maintenance of gut homeostasis. The authors also claim that attenuation of CTLA4 on ILCs activates type 1 immunity, which in turn leads to further gut inflammation. The observations made in the manuscript are interesting in terms of unveiling the role of CTLA4 induced in ILCs and might point towards one of the possible mechanisms of CPI-induced colitis. However, although the bioinformatic analysis was well performed, the overall phenotypes of genetic deletion or antibody blockade of Ctla4 in the innate immune compartment are subtle and thus the results need to be carefully interpreted to determine the impact of CTLA4 induction on ILCs during intestinal inflammation. In addition, although it may be beyond the scope of the current manuscript, the mechanism by which CTLA4 expression on ILCs lacking TCR results in the inhibitory phenotype remains unclear. Therefore, additional experiments should be performed to establish more solid evidence to support the authors' claims.

Major notes:

1. The phenotypes of Rag2 KO genetically deleted with Ctla4 (Fig4F-H) or treated with CPI (Fig6G-K) are subtle. Histological evaluations, and if possible, DAI scores (or any other phenotypic descriptions that authors can present) in both experiments are needed to show the impact of CTLA4-expressing ILCs on attenuating the colitis.
2. In the current manuscript, the abbreviation ‘FMT’ suddenly appears in Figure 6F-K without any explanation in the main text. It is not at all clear why TRUC mice derived FMT in combination with CPI was performed rather than CPI alone. Is it because the gut microbes from TRUC mice are transmissible and colitogenic? The authors need to explain why this model was used for the analysis.
3. In Fig6L, the authors show that ILC1 is elevated in human PBMCs harvested from CPI-colitis patients. Did the authors observe the qualitative changes (alteration in transcriptional signatures) in ILCs derived from patients with CPI enteritis? These experiments are required to translate mouse results into human biology.
4. In the abstract p3 line 41 ‘the CTLA4 expression is regulated by ILC-subset-specific cytokine cues in a microbiota-dependent manner.’ The data related to the gut microbiota are depicted only in Fig. 3 which shows the comparison of CTLA4 expression between SPF and antibiotic-treated/GF mice. In order to make such a statement, it is necessary to clarify through which pathways microbial signals regulate CTLA4 expression. Otherwise, the authors should tone down the statement.
5. It would be ideal to show the expression of CTLA4 in other immune cells in comparison with ILCs, maybe in Supplementary, in order to interpret the Rag2 KO Ctla4 KO results.

Minor note:

If Lin+CD127+ in Fig 5B-C indicates ILCs, this should be Lin-CD127+. Please check.

Reviewer #3:

Remarks to the Author:

In this study, Lo et al investigated whether ILC express CTLA4 and its role in the regulation of colitis and IBD. The first part of the study found different types of ILCs expressed CTLA4, which is interesting. However, the second part regarding the role of ILC expression of CTLA4 in colitis was very weak, which did not support the authors' conclusion that ILC expression of CTLA4 inhibits colitis or checkpoint colitis. Actually, the data presented in this study that Rag^{-/-}-CTLA4^{-/-} mice, unlike germ-line Ctla4^{-/-} mice, which develop a spontaneous and fatal phenotype, did not show any phenotype, indicating that innate expression of CTLA4 plays a minimal role, if any, in regulating colitis, which is against the author main conclusion. Even in DSS colitis, the severity of colitis was only moderately increased. Several statements in the manuscript were misleading, or overstated at the best. For examples, Tbx21^{-/-} x Rag2^{-/-} Ulcerative Colitis (TRUC) and anti-CD40 colitis in Rag^{-/-} mice are ILC-dependent (page 10) without literature support. Some other major concerns are needed to be addressed:

- 1) please modify the title as no data supporting "Microbiota-dependent expression of CTLA-4 by innate lymphoid cells restrains IFN γ dependent colitis"
- 2) The authors cited ref 63 from their early work which is still under review, which should not be cited as it has not been published. seems some data from ref 63 had also been presented in this manuscript (Figs 1 and 2, Supple Fig 1). If you wanted to present those data in this manuscript, please described how much they are overlapped and provide the deposit# of the scRNA-seq data to assist the review process. Please also describe the model used.
- 3) Fig4 basically showed the effect innate deficiency of CTLA4 on gene expression or DSS colitis, which are not ILC specific. If you wanted to make conclusion ILC-CTLA4, a ILC specific CTLA4 KO mice are required. Otherwise, from data of Fig4, you can not make such conclusion. Unless you can show other innate cells absolutely do not express CTLA4.
- 4) Supple Fig 3, 4.7% of IL-10 expression is not very few as stated. Please modify.
- 5) Page 11, please modify the conclusion "Together, these data indicate that when CTLA-4 is deficient from ILCs, there is a transcriptional drive towards a more pro-inflammatory IFN γ -mediated transcriptional profile". please change "when CTLA-4 is deficient from ILCs" to "when CTLA-4 is deficient from innate cells".
- 6) Fig 4 DSS colitis. Please provide histopathology, pathological scores, and proinflammatory cytokines data. Only weight change, which was very minimal, and colon mass are not enough to show the severity of colitis.
- 7) The last result part "ILC1 are expanded and mediate CPI-induced colitis". although interesting, but seems nothing supporting the main conclusion of the manuscript.
- 8) page 21. scRNA-seq. Please show the gating strategy for isolating CD45⁺ LP cells
- 9) Human samples. Please provide detail information of the patients in a table

REVIEWER COMMENTS

Reviewer #1 (Remarks to the Author):

Lo and colleagues report here that innate lymphoid cells (ILC) express the cytotoxic T lymphocyte-associated antigen 4 (CTLA-4) checkpoint with predominant expression in ILC1 and ILC3 regulated by cytokine cues and the microbiota. The genetic deletion of CTLA-4 in mice or blocking CTLA-4 with antibodies indicated the relevance of CTLA-4 expressed by ILCs for maintaining the integrity of intestinal homeostasis in chemical and immune checkpoint inhibitor (CPI) colitis models. These results are interesting considering these results and the clinical relevance of CPI colitis in tumor patients treated with checkpoint inhibitors.

We thank the reviewer for their comments and their interest in our work on ILCs expressing CTLA4 in the context of colitis.

Previous reports (PMID: 9430233) showed that the lymphoproliferative disorder reported in *Ctla4*^{-/-} mice is mediated by CD4 T cells as indicated in experiments with depletion of CD4 T cells in *Ctla4*^{-/-} animals. The reported difference in the percent of initial body weight loss between *Rag2*^{-/-} and *Rag2*^{-/-} x *Ctla4*^{-/-} challenged with Dextran Sodium Sulfate (DSS) is minor and not significant except on day 2. These findings question the relevance of ILC2 expressing CTLA-4 for developing the lymphoproliferative disorders observed in CTLA-4^{-/-} mice and potentially their relevance to CPI colitis.

Response: We thank the reviewer for their comments; however, we would like to clarify that we have not suggested that ILC-expressed CTLA-4 contributes to the lymphoproliferative disorder observed in CTLA-4^{-/-} mice. As the reviewer highlights, this is known to be mediated by loss of CTLA-4 on T cells, especially T_{regs}. In the colitis disease models used in our study, we have examined the potential roles for CTLA-4 expression by colonic ILCs, deliberately in the absence of T cells, by using *Rag2*^{-/-} mice, to enable us to circumvent the effects of these fatal lymphoproliferative disorders which would otherwise prevent analysis of CTLA-4 functions by ILCs. As shown in Supplementary Figure 10a and b, these *Rag2*^{-/-} x *Ctla4*^{-/-} do not develop spontaneous lymphoproliferative disorder. Given the large number of T cells expressing very high levels of CTLA-4 in WT mice, we reasoned this strategy to be superior to generation of new mouse lines with which to conditionally delete CTLA-4 on ILCs only, enabling us to examine ILC-dependent effects of CTLA-4 more specifically, without the potential for pathway redundancy, potentially introduced by the contaminating effects of CTLA-4 expressing T cells.

We agree that the difference in the two groups of mice is minor when treated with DSS. However, an updated analysis of the DSS experiment showed significance on day 3, as well as day 2 for weight loss and this has been amended in the revised manuscript (Supplementary Figure 15a). Additionally, for fuller characterisation of disease effects, we

have now included data on spleen mass and colon length (Supplementary Figure 15c and d), although these showed no differences between experimental groups. However, a significant increase in IFN γ production was evident in the absence of CTLA-4 (Supplementary Figure 15f), supporting the conclusion that CTLA-4 on ILCs may act to inhibit excessive ILC responses during disease and this elevated expression of IFN γ likely contributed to the small, but significantly greater early weight loss observed in DSS treated *Rag*^{-/-} x *Ctla4*^{-/-} mice relative to *Rag*^{-/-} animals.

To further substantiate our claims, we have now conducted TRUC (Tbx21^{-/-} x *Rag*^{-/-} mice in possession of an endogenous colitogenic microbiota) and anti-CD40 colitis models to the revised manuscript, which both enable assessment of innate cell contribution to colitis disease processes, with immunopathology previously demonstrated to be ILC-driven in both cases by Powell et al and Uhlig et al. In both instances, when CTLA-4 activity was impeded (using anti-CTLA-4 antibodies in TRUC mice, and through genetic ablation using *Rag*^{-/-} x *Ctla4*^{-/-} mice in the case of anti-CD40 colitis induction) clinical phenotypes with more severe disease were observed. TRUC mice treated with anti-CTLA-4 exhibited significantly lower survival, greater colon and spleen mass and significantly increased neutrophilic influx into colonic tissues (Figures 4d-f and Supplementary Figure 13b) and anti-CD40 treated *Ctla4* deficient mice demonstrated significantly greater peak weight loss day 4 post treatment, greater spleen mass and trends towards lower survival rates and more severe disease severity index (4g, 4i and Supplementary figures 14a, 14b).

Therefore, we have now demonstrated using 3 independent colitis disease models that removal of CTLA-4 activity within the innate immune compartment contributes to worsened disease processes. We hope the reviewer agrees with us, that these additions significantly strengthen the conclusions drawn by our study, regarding identification of a role for ILC expression of CTLA-4 in regulating these aberrant disease phenotypes.

The authors report that the intestinal microbiota or the stimulation with a cytokine cocktail (IL-12 + IL-18 for Nkp46+ ILCs) induces *Ctla4* expression by ILCs as indicated by experiments with antibiotics-treated mice, germ-free animals, or in vitro stimulation. Information on how these different findings are linked to driving *Ctla4* expression by ILCs must be provided.

Response: We have now performed further microbiota related experiments, including addition of an ex-GF group (GF mice recolonised with SPF microbiota through 4 weeks of co-housing with SPF mice) to our analyses. In ex-GF mice, loss of ILC CTLA-4 expression observed in GF mice was restored to similar levels as SPF mice, indicating that presence of microbial cues is sufficient to drive CTLA-4 expression by ILCs (Figure 3a and b). Furthermore, ELISA analysis of the supernatant from in vitro cultured colonic biopsies from this experiment showed that IL12 was significantly produced in ex-GF mice and IL-18 was reduced in GF mice and this could be returned to SPF levels in the ex-GF mice (Supplementary Fig. 9b). These data provide additional support for our argument that the

presence of commensal microbial cues is critical for induction of specific cytokine signals which drive homeostatic ILC expression of CTLA-4.

These data enhance the *in vitro* data seen in Figure 2h and possible links the cytokine cues which drives Ctl4 expression on Nkp46⁺ ILCs. We also used a mix of proinflammatory microbiota (FMT), obtained from the TRUC colony and gavaged it into *Rag2*^{-/-} and *Rag2*^{-/-} *x* *Ctla4* mice and found that the only mice to have any clinical differences and increase in infiltrating neutrophils was in the *Rag2*^{-/-} *x* *Ctla4* mice treated with FMT (Figure 3c-e). Moreover, the only cytokine differences seen were in both IFN γ producing ILC1s and Nkp46⁺ ILC3s (Figure 3f-g). These data from the FMT treated *Rag2*^{-/-} *x* *Ctla4* again highlighted that a proinflammatory microbiota could drive an inflammatory ILC response when Ctl4 is missing.

The authors need to suggest mechanisms on how CTLA-4 restricts the suggested immunophenotype mediated by ILCs in DSS and CPI colitis. The manuscript focuses on exacerbated IFN γ production, but the manuscript does not touch other potential intrinsic and extrinsic pathways, such as ligand competition, inhibitory signaling, or modulating the function of antigen-presenting cells by reverse signaling.

Response: We thank the reviewer for their comment and agree this is an interesting question. In T cell biology, it is known that CTLA-4 competes with its homologue CD28 for binding to their shared ligands (CD80/CD86), with CTLA-4 having a higher CD80/86 affinity. In response to the reviewer's comment, we have therefore investigated whether CD28 is expressed on ILC populations, as a first step to understanding whether CTLA-4 might function similarly in ILCs and T cells. We found that CD28 could be readily detected on intestinal ILCs, and that its expression predominantly mapped to the same ILC subsets that expressed CTLA-4 (Nkp46⁺ ILC1s and ILC3s) (Supplementary Figure 6). To test for a potential role for CD28 in ILC biology, we generated and characterised a new *Rag2*^{-/-} *x* *Cd28*^{-/-} mouse line, allowing us to study CD28 function in the absence of T cells (Supplementary Figure 7-8). Strikingly, we found that deletion of CD28 in *Rag2*^{-/-} mice resulted in a significant loss of ILC1s, ILC2s and Nkp46⁺ ILC3s in the colon, with the remaining cells showing phenotypic alterations including increased CTLA-4 expression and IFN γ production. We hypothesise that CD28 plays a key role in maintaining colonic ILCs and its loss perturbs intestinal homeostasis leading to ILC activation and increased colon mass (Supplementary Figure 7c). While dissecting this interplay further is beyond the scope of the current manuscript, the demonstration that CD28 is expressed on ILCs and can modulate their homeostasis and function suggests CTLA-4 has the potential to alter ILC biology by regulating CD28 signalling pathways.

Moreover, the methods could be more precisely described. Mainly the CPI colitis model needs to be explained. The authors have submitted a second manuscript describing the CPI colitis model. Still, I need access to this manuscript to explain the CPI model in detail to review its significance of the CPI model for this manuscript.

Response: We apologise for the lack of clarity regarding the reporting of methodology in our study. In our revised manuscript, we have taken care to improve the details for methods used, and a schematic of the experimental plan of the CPI-C model in *Rag*^{-/-} mice has been provided (Supplementary Figure 19a). Furthermore, the CPI-C mouse model manuscript has now been published and is fully cited in our revised work (citation no. 64 (Lo et al. PMID: 37872166)).

Major points

1. Page 6, lines 126 to 129, describing figure 1, panel D. It remains to be seen in the text that these results have been derived from the Krzywinska data set. Moreover, figure 1 focuses on small intestinal *Ctla4* expression by ILCs, whereas most of the manuscript focuses on the large intestine. Authors could, for example, also analyze the GSE150050 datasets to show CTLA-4 expression by human colonic ILCs.

Response: We thank the reviewer for their useful suggestions. In response, we have provided a more detailed description for our use of the Krzywinska dataset in the manuscript text. Furthermore, at the reviewer's suggestion, we have included an analysis of the GSE150050 (Mazzurana) dataset (Supplementary Figure 16). However, an issue with the use of the Mazzurana dataset for deriving useful conclusions from, is the extremely small size of the annotated ILC1 and ILC2 clusters, limiting its usefulness for enquiry into ILC3 only. Notably, the authors of the dataset outline in their study that the reason for this absence of ILC1/ILC2 is because the tissues used for these analyses derived from healthy individuals. Likely, this also underlies the reason why our analysis of this dataset did not reveal CTLA-4 expression in ILC3, as the expression of CTLA-4 in healthy controls at both transcript and protein levels were very low in our own patient analysis (Figure 5).

2. Presenting the representative flow cytometry plots for all analyzed ILC subtypes is more interesting. Consider moving the dot plots of the supplementary figure 2, panel A, to the central figure. Figure 2, panels A and B, the *Rag2*^{-/-} x *Ctla4*^{-/-} animals serve as a control. I do not think that in all summary plots, the negative control has to be presented. The corresponding figure legends do not describe what population was gated for the representative dot plots of panel A.

Response: Supplementary Figure 2 has now replaced Figure 2A and the summary dot plots for Figure 2B only show the CTLA-4 expressing cells in *Rag*^{-/-} mice. Furthermore, the gating strategy in Figure 2A has now highlighted which ILC population is described for Figure 2B.

3. In the figure legend of Figure 2, panel C, please state that the WT animals are BALB/c mice. On page 8, line 157, please explain why BALB/c mice have been used as WT animals.

Add the C57BL/6 WT controls to the panels with the Rag2^{-/-} and RORγt-eGFP mice. Response: We apologise for the previous omission of these details. Full strain details have now been added to both the main body of the text and the figure legends. Wildtype animals were selected to match the comparator strains: the Rag2^{-/-} and Rag2^{-/-} x Ctla4^{-/-} mice used in Figure 2 were BALB/c background, and therefore BALB/c wildtype controls were utilised.

4. Figure 2, panels C to H, there is also the possibility that the Ctla-4 homolog CD28 is upregulated by PMA/Iono or cytokine cocktail stimulation. It might also mean that activated ILCs and not resting ILCs express Ctla-4, which could be further emphasized by detecting other activation markers.

Response: We appreciate the reviewer's thoughts on this topic and agree that CTLA-4 induction is most likely linked to ILC activation. Consistent with this, the forward scatter profile of CTLA-4-positive ILC1s suggests they are larger than their CTLA-4-negative counterparts (Shown below in Response Figure A).

Response Figure A. Representative flow plot showing larger forward and side scatter in CTLA-4⁺ ILCs compared to CTLA-4⁻ ILCs from Balb/C wildtype mice

In T cells, TCR signalling is the major driver of CTLA-4 upregulation, however we contend that subset-specific cytokines co-opt this role in ILCs (Figure 2 g,h). Regarding CD28 expression, we have now examined this point and find that, unlike CTLA-4, ILC CD28 expression does not appear to increase upon PMA/Iono stimulation (Response Figure B). This opens up the possibility for different activation signals to alter the balance between CTLA-4 and CD28 expression on ILCs, presenting an interesting avenue for future investigation.

Response Figure B. Representative flow plot showing CD28 expression in unstimulated lineage⁻ IL-7R⁺ ILCs compared to PMA and ionomycin stimulated lineage⁻ IL-7R⁺ ILCs from the colonic lamina propria of BALB/c wildtype mice.

5. Please indicate on page 9, line 193, that WT animals are C57BL/6 mice.

Response: This has now been updated

6. The presented results of Figure 3 show that ILCs of antibiotic-treated and germ-free animals have reduced Ctla-4 expression. Since the literature is controversial if the number of ILCs is altered in germ-free and antibiotic-treated animals, indicate whether in these animals the ILC numbers are reduced or not changed. In addition, colonize germ-free animals with a microbiota to show that the microbiota upregulates Ctla-4, as stated on page 10, line 208. In the same experiment, please measure IL-12 and IL-18 to learn whether IL-12 and IL-18 mediate the effects of the microbiota on Ctla-4 expression by ILCs.

Response: We thank the reviewer for their suggestions. In response, the two papers described have now been cited in the discussion (citation 79 and 82) with how our finding on ILC absolute numbers differs to theirs. We have also included the ILC numbers in Supplementary Figure 9a. The proposed recolonisation of SPF microbiota into GF mice has also been performed with the finding that CTLA-4 expression is restored in Figure 3a and b. We have also added in ELISA data for IL-12 and IL-18 from these SPF, GF, and ex-GF mice (Supplementary Figure 9b) to show their changes in response to modulation of the presence of microbial cues. We hope the reviewer finds these additions satisfactory.

7. Since the reported differences in the percent of initial body weight loss between Rag2^{-/-} and Rag2^{-/-} x Ctla4^{-/-} exposed to Dextran Sodium Sulfate (DSS) are small, additional read-out parameters, such as histology, disease activity scores, and colon length, are warranted.

Response: We agree with the reviewer and have included colon length and spleen mass as additional readouts. As the changes in disease were small in this model, we have moved these data to revised Supplementary Figure 15. Instead, we have included new data for

anti-CD40 and TRUC disease models in the main figures which demonstrated more compelling disease readouts.

8. Figure 5, the presented heat map of an RNA-seq experiment of panel A, remains somewhat unclear if the submitted data are obtained from the re-analysis of a previous data set or whether this data stems from a newly generated data set. The IBD characteristics, including disease activity and course and immunosuppressive therapies, from which the material for the presented RNA-seq and flow cytometry data stems, need to be given. Also, the authors may consider offering Ctl4 expression from the non-inflamed regions of the same patients to exclude the possibility that individual differences and batch effects may have contributed to the reported results.

Response: We apologise for this lack of clarity. This bulk RNA-seq data is our own new dataset and will be repositied on GEO to be made available to the public following publication of our study. To make this clearer, text has now been added at the beginning of the results section introducing Figure 5 and in the methodology reporting for the human samples.

The requested patient metadata for Figure 5 have also been added to revised Supplementary Table 1. Notably, these patients were not being treated with thiopurines or biologics and only 2 were on steroids. Whilst we agree that this would have been an ideal control, it was not accommodated within our ethics, and it is now not possible to get non-inflamed regions from these patients. Therefore, the use of healthy controls remains the best available control group option.

9. In Figure 6, panels F to J, the 'FMT' is not explained in the text describing the figure. The authors may also consider re-analyzing the GSE144469 data set for CTLA-4 expression by colonic ILCs in addition to the presented flow cytometry data from the peripheral blood.

Response: We apologise for this oversight and have now added more detail in the text regarding the 'faecal microbial transplantation' (FMT) model, citing also our (now published) CPI-C paper (Lo *et al.*, 2023, citation no. 64. PMID: 37872166). Furthermore, we have included an experimental plan schematic for the CPI-C induction in *Rag2*^{-/-} mice (Supplementary Figure 19a).

The analysis of CPI-C patient peripheral blood has been taken out of the revised manuscript, however an analysis of the Luoma *et al.*, dataset (GSE144469) has now been included (Supplementary Figure 20). We found that the normalized expression of CTLA-4 is relatively low in these two ILC clusters and this is most likely due to the limitations of the experiment and 10x single cell sequencing when running bulk CD45+ sorted cells. Furthermore, normalising to all clusters greatly reduces the expression of CTLA-4 on ILCs due to their lower expression in comparison to the T cell clusters identified by Luoma *et al.* However, we were able to show an expansion in their identified ILC1 cluster when

comparing cell abundance changes and also gene, and pathway analysis demonstrate an increase in interferon related genes, as discussed in the manuscript revised text.

10. Methods

- Animal husbandry: The animal protocol number of the local animal welfare committee missing

Response: We apologise for this oversight. This has been added to the methods including PPL numbers and the review of the AWERB and Home Office and is stated as: "All procedures were conducted under licenses (Home Office Licence Numbers PPL: 70/6792, 70/8127, 70/7869, P8999BD42) from the United Kingdom (UK) Home Office in accordance with The Animals (Scientific Procedures) Act 1986 and licences were approved by each Animal Welfare and Ethical Review Body."

- Colitis models: Have littermates been used?

Response: Littermate controls have been used for all the colitis models except those involving *Rag2*^{-/-} x *Ctla4*^{-/-} mice. The latter were maintained as a homozygous line and were compared with the parental *Rag2*^{-/-} strain bred in the same facility, matching for age and sex.

-Flow cytometry: clone numbers of the used antibodies missing

Response: Antibody details have now been included in revised Supplementary Table 2

- Antibiotic treatment: Since the antibiotics have been given in the drinking water, each animal's dose is difficult to judge.

Response: While we agree with the reviewer's point, administration of antibiotics in drinking water is a very widely and commonly applied methodology for the depletion of the gut microbiota. Furthermore, evidence in literature indicates that administration of antibiotics in drinking water is a successful and efficient method for depletion of gut bacteria, and may be superior to administration by oral gavage once daily, as this method can occasionally result in the biased hyperproliferation of some taxons, such as the Gammaproteobacteria (PMID: 33176677).

- Immune checkpoint colitis not described.

Response: These details have been added to the pre-clinical models of colitis and our recently published CPI-C model paper has now been cited too (Lo *et al.*, 2023, citation no. 64. PMID: 37872166). We have also added an experimental plan schematic for the CPI-C induction in these *Rag2*^{-/-} mice in Supplementary Figure 19a.

Minor

1. The clusters are not annotated in the UMAP graph, panel A, supplementary figure 1.

Response: This has been amended now.

2. Flow cytometry dot blots: The manuscript does not give x- and y-axis scales.

Response: This has been amended now.

3. On page 8, line 165, correct the spelling mistake "non-soecific".

Response: This has been amended now.

Reviewer #2 (Remarks to the Author):

In this manuscript, Lo and colleagues describe the induction of CTLA4 on ILC1/3 driven by gut microbial and inflammatory cues and its involvement in the maintenance of gut homeostasis. The authors also claim that attenuation of CTLA4 on ILCs activates type 1 immunity, which in turn leads to further gut inflammation. The observations made in the manuscript are interesting in terms of unveiling the role of CTLA4 induced in ILCs and might point towards one of the possible mechanisms of CPI-induced colitis. However, although the bioinformatic analysis was well performed, the overall phenotypes of genetic deletion or antibody blockade of *Ctla4* in the innate immune compartment are subtle and thus the results need to be carefully interpreted to determine the impact of CTLA4 induction on ILCs during intestinal inflammation. In addition, although it may be beyond the scope of the current manuscript, the mechanism by which CTLA4 expression on ILCs lacking TCR results in the inhibitory phenotype remains unclear. Therefore, additional experiments should be performed to establish more solid evidence to support the authors' claims.

Response: We thank the reviewer for their comments and their interest in the data in this manuscript. We believe the extra data and experiments we have presented in our revised work now help to answer these points and strengthen the support for the claims made throughout our study.

Major notes:

1. The phenotypes of Rag2 KO genetically deleted with *Ctla4* (Fig4F-H) or treated with CPI (Fig6G-K) are subtle. Histological evaluations, and if possible, DAI scores (or any other phenotypic descriptions that authors can present) in both experiments are needed to show the impact of CTLA4-expressing ILCs on attenuating the colitis.

Response: We agree that the difference in the two groups of mice is minor when treated with DSS. However, an updated analysis of the DSS experiment showed significance on day 3, as well as day 2 and this has been amended in the revised manuscript (Supplementary Figure 15a). Additionally, for fuller characterisation of disease effects, we

have now included data on spleen mass and colon length, although these showed no differences between experimental groups. However, a significant increase in IFN γ production was evident in the absence of CTLA-4, supporting the conclusion that CTLA-4 on ILCs may act to inhibit excessive ILC responses during disease and this elevated expression of IFN γ likely contributed to the small, but significantly greater early weight loss observed in DSS treated *Rag*^{-/-} x *Ctla4*^{-/-} mice relative to *Rag*^{-/-} animals.

To further substantiate our claims, we have now conducted TRUC (Tbx21^{-/-} x *Rag*^{-/-} mice in possession of an endogenous colitogenic microbiota) and anti-CD40 colitis models to the revised manuscript, which both enable assessment of innate cell contribution to colitis disease processes, with immunopathology previously demonstrated to be ILC-driven in both cases (Powell et al 2012 and Uhlig et al 2006). In both instances, when CTLA-4 activity was impeded (using anti-CTLA-4 antibodies in TRUC mice, and through genetic ablation using *Rag2*^{-/-} x *Ctla4*^{-/-} mice in the case of anti-CD40 colitis induction) clinical phenotypes with more severe disease were observed. TRUC mice treated with anti-CTLA-4 exhibited significantly lower survival, greater colon and spleen mass and significantly increased neutrophilic influx into colonic tissues (Figures 4d-f and Supplementary Figure 13b) and anti-CD40 treated *Ctla4* deficient mice demonstrated significantly greater peak weight loss day 4 post treatment, greater spleen mass and trends towards lower survival rates and more severe disease severity index (Figures 4g, 4i and Supplementary figures 14a, 13b).

Therefore, we have now demonstrated using 3 independent colitis disease models that removal of CTLA-4 activity within the innate immune compartment contributes to worsened disease processes. We hope the reviewer agrees with us, that these additions significantly strengthen the conclusions drawn by our study, regarding identification of a role for ILC expression of CTLA-4 in regulating these aberrant disease phenotypes.

2. In the current manuscript, the abbreviation 'FMT' suddenly appears in Figure 6F-K without any explanation in the main text. It is not at all clear why TRUC mice derived FMT in combination with CPI was performed rather than CPI alone. Is it because the gut microbes from TRUC mice are transmissible and colitogenic? The authors need to explain why this model was used for the analysis.

Response: We apologise for this oversight and have now added more detail in the text regarding the 'faecal microbial transplantation' (FMT) model (the first time it is mentioned now in the new Figure 3c) with the phrase "an oral gavage of known pro-inflammatory colitogenic inducing microbiota from TRUC mice (FMT)⁷²" (where reference 72 is the original TRUC paper by Garrett et al.), citing also our (now published) CPI-C paper (Lo et al., 2023, citation no. 64. PMID: [37872166](https://pubmed.ncbi.nlm.nih.gov/37872166/)) from Figure 6 onwards. Furthermore, we have included an experimental plan schematic for the CPI-C induction in *Rag2*^{-/-} mice (Supplementary Figure 19a) and gone into more depth in the methods section

As demonstrated in Lo *et al.*, 2023, FMT alone or combination CPI treatment alone is not sufficient to induce colitis in wildtype mice, instead requiring both treatments to be given to induce disease. In our manuscript, the *Rag2*^{-/-} mice, mice treated with FMT alone had slightly higher colon and spleen mass, but these were not significant and neutrophil infiltration was unaffected (Figures 6g-i). However combinatorial FMT and CPI treatment did significantly increase these clinical parameters, supporting the notion that innate cell derived CTLA-4 acts to restrict ILC functionality in response to inflammatory stimuli. As the reviewer correctly notes, TRUC derived FMT functions due to our previous observation that the microbiota harboured by TRUC mice is transmissible and colitogenic (PMID: 23063332), but this is only found within TRUC mice and TRnUC mice (mice which lack the microbiota) when they are both co-housed together. *Rag* mice inoculated with *Helicobacter Typhlonius* have been shown to not develop colitis from our previous observations (PMID: 23063332) showing that the gut microbes are only transmissible between mice lacking both T-bet and *Rag* genes.

3. In Fig6L, the authors show that ILC1 is elevated in human PBMCs harvested from CPI-colitis patients. Did the authors observe the qualitative changes (alteration in transcriptional signatures) in ILCs derived from patients with CPI enteritis? These experiments are required to translate mouse results into human biology.

Response: During the process of revising our manuscript, we have now excluded these data, due to the patient samples being from peripheral blood PBMC, which we ultimately felt may lack relevance for analysis of tissue-specific immune cell dynamics (PMID: 34035530). We have replaced this dataset with a reanalysis of a single cell RNA-seq data from patients with CPI-Colitis (Luoma *et al.*, dataset GSE144469) in which they identified two clusters of colonic ILCs (Supplementary Figure 20). These data show similar transcriptional changes between healthy and CPI-C patients, with an expansion in the ILC1 cluster and have an increase in IFN γ related genes and genes associated with the IFN γ response pathway, similar to our mouse CPI-C model in wildtype mice (Figure 6a-d). We hope the reviewer agrees with the rationale for these decisions and finds the inclusion of these new data to be beneficial to our revised work.

4. In the abstract p3 line 41 'the CTLA4 expression is regulated by ILC-subset-specific cytokine cues in a microbiota-dependent manner.' The data related to the gut microbiota are depicted only in Fig. 3 which shows the comparison of CTLA4 expression between SPF and antibiotic-treated/GF mice. In order to make such a statement, it is necessary to clarify through which pathways microbial signals regulate CTLA4 expression. Otherwise, the authors should tone down the statement.

Response: We have now performed further microbiota related experiments, including addition of an ex-GF group (GF mice recolonised with SPF microbiota through 4 weeks of co-housing with SPF mice) to our analyses. In ex-GF mice, loss of ILC CTLA-4 expression

observed in GF mice was restored to similar levels as SPF mice, indicating that presence of microbial cues is sufficient to drive CTLA-4 expression by ILCs (Figure 3a and b). Furthermore, ELISA analysis of the supernatant from *ex vitro* cultured colonic biopsies demonstrated that IL-18 was significantly reduced in GF mice and this could be returned to SPF levels following microbial colonisation in the ex-GF mice. Similarly, ex-GF mice also had elevated levels of colonic IL-12 production (Supplementary Fig. 9b). These data provide additional support for our argument that the presence of commensal microbial cues are critical for induction of specific cytokine signals which drive homeostatic ILC expression of CTLA-4.

5. It would be ideal to show the expression of CTLA4 in other immune cells in comparison with ILCs, maybe in Supplementary, in order to interpret the Rag2 KO Ctl4 KO results.

Response: We thank the reviewer for their suggestion. These extra experimental data have now been performed and included in the manuscript (Supplementary Figure 3). We used wildtype mice to compare CD4⁺ and CD11b⁺ cells off the gating of the lineage⁺ cells and to compare the expression of CTLA-4 with the identified ILC subsets and shows that the CTLA-4 expressing ILCs are robust.

Minor note:

If Lin⁺CD127⁺ in Fig 5B-C indicates ILCs, this should be Lin⁻CD127⁺. Please check.

Response: We apologise for the lack of clarity. The Lin⁺ CD127⁺ plots and summary dot plots shown here were used as a positive control of bulk T cells (Supplementary Figure 17a for the gating strategy) for the subsequent ILC subsets and their CTLA-4 expression. Therefore, although this is not a mistake, a sentence has been added to the revised text to make this clearer: "We found that CTLA-4 was present in human ILC1 and NCR⁻ ILC3 cells, when comparing Lineage⁺ IL-7R⁺ shown as a positive control for CTLA-4 staining on bulk T cells (Fig. 5b and c and Supplementary Fig. 17a)"

Reviewer #3 (Remarks to the Author):

In this study, Lo et al investigated whether ILC express CTLA4 and its role in the regulation of colitis and IBD. The first part of the study found different types of ILCs expressed CTLA4, which is interesting. However, the second part regarding the role of ILC expression of CTLA4 in colitis was very weak, which did not support the authors' conclusion that ILC expression of CTLA4 inhibits colitis or checkpoint colitis. Actually, the data presented in this study that Rag^{-/-}CTLA4^{-/-} mice, unlike germ-line Ctl4^{-/-} mice, which develop a spontaneous and fatal phenotype, did not show any phenotype, indicating that innate expression of CTLA4 plays a minimal role, if any, in regulating colitis, which is against the author main conclusion. Even in DSS colitis, the severity of colitis was only moderately

increased. Several statements in the manuscript were misleading, or overstated at the best. For examples, *Tbx21*^{-/-} x *Rag2*^{-/-} Ulcerative Colitis (TRUC) and anti-CD40 colitis in *Rag*^{-/-} mice are ILC-dependent (page 10) without literature support. Some other major concerns are needed to be addressed:

Response: We agree that the difference in the two groups of mice is minor when treated with DSS. However, an updated analysis of the DSS experiment showed significance on day 3, as well as day 2 for weight loss, and this has been amended in the revised manuscript (Supplementary Figure 15a). Additionally, for fuller characterisation of disease effects, we have now included data on spleen mass and colon length, although these showed no differences between experimental groups. However, a significant increase in IFN γ production was evident in the absence of CTLA-4, supporting the conclusion that CTLA-4 on ILCs may act to inhibit excessive ILC responses during disease and this elevated expression of IFN γ likely contributed to the small, but significantly greater early weight loss observed in DSS treated *Rag*^{-/-} x *Ctla4*^{-/-} mice relative to *Rag*^{-/-} animals.

To further substantiate our claims, we have now conducted TRUC (*Tbx21*^{-/-} x *Rag*^{-/-} mice in possession of an endogenous colitogenic microbiota) and anti-CD40 colitis models to the revised manuscript, which both enable assessment of innate cell contribution to colitis disease processes, with immunopathology previously demonstrated to be ILC-driven in both cases (Powell et al and Uhlig et al have now both been cited within the manuscript).

In both instances, when CTLA-4 activity was impeded (using anti-CTLA-4 antibodies in TRUC mice, and through genetic ablation using *Rag*^{-/-} *Ctla4*^{-/-} mice in the case of anti-CD40 colitis induction) clinical phenotypes with more severe disease were observed. TRUC mice treated with anti-CTLA-4 exhibited significantly lower survival, greater colon and spleen mass and significantly increased neutrophilic influx into colonic tissues (Figures 4d-f and Supplementary Figure 13b) and anti-CD40 treated *Ctla4* deficient mice demonstrated significantly greater peak weight loss day 4 post treatment, greater spleen mass and trends towards lower survival rates and more severe disease severity index (Figures 4g, 4i and Supplementary figures 14a, 14b).

Therefore, we have now demonstrated using 3 independent colitis disease models that removal of CTLA-4 activity within the innate immune compartment contributes to worsened disease processes. We hope the reviewer agrees with us, that these additions significantly strengthen the conclusions drawn by our study, regarding identification of a role for ILC expression of CTLA-4 in regulating these aberrant disease phenotypes.

1) please modify the title as no data supporting "Microbiota-dependent expression of CTLA-4 by innate lymphoid cells restrains IFN γ dependent colitis"

Response: At the reviewer's suggestion, we have altered the title, which now reads: "CTLA-4 expressing innate lymphoid cells modulate mucosal homeostasis in a microbiota dependent manner"

However, we have chosen to retain the conclusion that CTLA-4 expression is microbiota-dependent, based on additional experiments which we have performed and presented in the revised work. This includes addition of an ex-GF group (GF mice recolonised with SPF microbiota through 4 weeks of co-housing with SPF mice) to our analyses. In ex-GF mice, loss of ILC CTLA-4 expression observed in GF mice was restored to similar levels as SPF mice, indicating that presence of microbial cues is sufficient to drive CTLA-4 expression by ILCs (Figure 3a and b). Furthermore, ELISA analysis of the supernatant from *ex vitro* cultured colonic biopsies demonstrated that IL-18 was significantly reduced in GF mice and this could be returned to SPF levels following microbial colonisation in the ex-GF mice. Similarly, ex-GF mice also had elevated levels of colonic IL-12 production (Supplementary Fig. 9b).

These data provide additional support for our thesis, that the presence of commensal microbial cues are critical for induction of specific cytokine signals which drive homeostatic ILC expression of CTLA-4. We hope the reviewer agrees with the inclusion of this new data and its value in further supporting the conclusions we have drawn.

2) The authors cited ref 63 from their early work which is still under review, which should not be cited as it has not been published. It seems some data from ref 63 had also been presented in this manuscript (Figs 1 and 2, Supple Fig 1). If you wanted to present those data in this manuscript, please describe how much they are overlapped and provide the deposit# of the scRNA-seq data to assist the review process. Please also describe the model used.

Response: We apologise for this oversight. Previous Ref 63, which is now 64 (Lo et al., 2023) is now published (PMID: [37872166](https://pubmed.ncbi.nlm.nih.gov/37872166/)). The GEO accession code for figure 6 (GSE222959 "Single cell transcriptomics reveals colonic lymphocyte remodelling and emergence of polyfunctional, cytolytic lymphocyte responses in CPI-induced colitis") is provided in the Data availability section. However, none of the analysis in this manuscript is included in Lo et al., 2023, which does not look at the ILC clusters that we previously identified. The model is now also further described in the methods and results text, as well as including an experimental schematic in Supplementary Figure 19a.

3) Fig4 basically showed the effect innate deficiency of CTLA4 on gene expression or DSS colitis, which are not ILC specific. If you wanted to make conclusion ILC-CTLA4, ILC specific CTLA4 KO mice are required. Otherwise, from data of Fig4, you can not make such conclusion. Unless you can show other innate cells absolutely do not express CTLA4.

Response: We appreciate the reviewer's concerns. Although generation of an ILC specific cre/flox system for deletion of *Ctla4*, with subsequent repetition of our colitis models is not feasible for us at this current time, given the large number of T cells expressing very high levels of CTLA-4 in WT mice, we reasoned the strategy we have undertaken to be a valuable alternative to generation of new mouse lines with which to conditionally delete

CTLA-4 on ILCs only. This is because it enables us to examine ILC-dependent effects of CTLA-4 more specifically, without the potential for pathway redundancy potentially introduced by the contaminating effects of CTLA-4 expressing T cells. Additionally, we have provided new data which utilises two innate models of colitis known to be driven by ILC-dependent activities (Powell et al 2012 and Uhlig et al 2006). These new data further support our conclusions that ILC-driven colitis severity is worsened in the absence of ILC-CTLA-4. Furthermore, re-analysis of identifiable cell types from our current flow cytometry data (Supplementary Figure 3) indicates detection of CTLA-4 on other innate cells such as CD11b cells was not possible – supporting the notion that helper-like ILCs are a dominant innate source of CTLA-4 in the colon. From our flow cytometry data, the lineage⁻ IL-7R⁻ Nkp46⁺ cells do not express CTLA-4 either showing that NK cells in these mice do not exist and make the data more convincing.

4) Supple Fig 3, 4.7% of IL-10 expression is not very few as stated. Please modify.

Response: This has now been modified and now reads: "We detected no IL-10-producing ILCs upon stimulation with PMA and ionomycin in the cLP and minimal (4.7%) IL-10-producing ILCs in the small intestine, but none of these co-expressed CTLA-4 (Supplementary Fig. 5a)"

5) Page 11, please modify the conclusion "Together, these data indicate that when CTLA-4 is deficient from ILCs, there is a transcriptional drive towards a more pro-inflammatory IFN γ -mediated transcriptional profile". please change "when CTLA-4 is deficient from ILCs" to "when CTLA-4 is deficient from innate cells".

Response: This has now been modified and now reads: "Together, these data indicate that when CTLA-4 is deficient from innate cells, there is a transcriptional drive towards a more pro-inflammatory IFN γ -mediated transcriptional profile."

6) Fig 4 DSS colitis. Please provide histopathology, pathological scores, and proinflammatory cytokines data. Only weight change, which was very minimal, and colon mass are not enough to show the severity of colitis.

Response: We agree with the reviewer, and have included other available data for these experiments, including colon length and spleen mass as additional readouts. As already discussed, because the changes in disease were small in this model, we have moved these data to revised Supplementary Figure 15. Instead, we have included new data for anti-CD40 and TRUC disease models in the main figures which demonstrated more compelling disease readouts.

7) The last result part "ILC1 are expanded and mediate CPI-induced colitis". although interesting, but seems nothing a=supporting the main conclusion of the manuscript.

Response: During the process of revising our manuscript, we have excluded these data, due to the patient samples being from peripheral blood PBMC which we ultimately felt may lack relevance for analysis of tissue-specific immune cell dynamics (PMID: 34035530). We have replaced this dataset with a reanalysis of a single cell RNA-seq data from patients with CPI-Colitis (Luoma *et al.*, dataset GSE144469) in which they identified two clusters of colonic ILCs (Supplemental Figure 20). These data show similar transcriptional changes between healthy and CPI-C patients, with an expansion in the ILC1 cluster and have an increase in IFN γ related genes and genes associated with the IFN γ response pathway, similar to our mouse CPI-C model in wildtype mice (Figure 6a-d). We hope the reviewer agrees with the rationale for these decisions and finds the inclusion of these new data to be beneficial to our revised work.

8) page 21. scRNA-seq. Please show the gating strategy for isolating CD45+ LP cells

Response: This has now been included in the manuscript on page 17 and Supplementary Fig. 18a

9) Human samples. Please provide detail information of the patients in a table

Response: We apologise for this oversight. The requested patient metadata for Figure 5 have also been added to revised Supplementary Table 1.

REVIEWERS' COMMENTS

Reviewer #1 (Remarks to the Author):

I thank the authors for their careful response to the reviewer's suggestions.

Reviewer #2 (Remarks to the Author):

The authors adequately responded to the reviewer's comments, and the manuscript is now improved.

Reviewer #3 (Remarks to the Author):

All my previous concerns have been addressed appropriately

Reviewer #1 (Remarks to the Author):

I thank the authors for their careful response to the reviewer's suggestions.

Response: We thank the reviewer for their time reviewing our manuscript.

Reviewer #2 (Remarks to the Author):

The authors adequately responded to the reviewer's comments, and the manuscript is now improved.

Response: We thank the reviewer for their time reviewing our manuscript.

Reviewer #3 (Remarks to the Author):

All my previous concerns have been addressed appropriately

Response: We thank the reviewer for their time reviewing our manuscript.